# A Critical Look at Targeted Instruction Selection:
# Disentangling What Matters (and What Doesn't)

Nihal V. Nayak [1]  Paula Rodriguez-Diaz [1]  Neha Hulkund [2]  Sara Beery [2]  David Alvarez-Melis [1 3]

## Abstract

Instruction fine-tuning of large language models (LLMs) often involves selecting a subset of instruction training data from a large candidate pool, using a small query set from the target task. Despite growing interest, the literature on targeted instruction selection remains fragmented and opaque: methods vary widely in selection budgets, often omit zero-shot baselines, and frequently entangle the contributions of key components. As a result, practitioners lack actionable guidance on selecting instructions for their target tasks. In this work, we aim to bring clarity to this landscape by disentangling and systematically analyzing the two core ingredients: data representation and selection algorithms. Our framework enables controlled comparisons across models, tasks, and budgets. We find that only gradient-based data representations choose subsets whose similarity to the query consistently predicts performance across datasets, models, and candidate pools. While no single method dominates, gradient-based representations paired with greedy round-robin selection often perform best on average at low budgets, but these gains diminish at larger budgets. Finally, we unify several existing selection algorithms as forms of approximate distance minimization between the selected subset and the query set, and support this view with new generalization bounds. More broadly, our findings provide critical insights and a foundation for more principled data selection in LLM fine-tuning. The code is available at https://github.com/dcml-lab/targeted-instruction-selection.

## 1. Introduction

Large language models (LLMs) have demonstrated a remarkable ability to follow complex instructions through instruction fine-tuning on curated datasets of instruction-response pairs (Hurst et al., 2024; Olmo et al., 2025; Agarwal et al., 2025). Constructing such datasets, however, typically requires careful experimentation and extensive ablation studies, making them both time-consuming and computationally expensive (Guha et al., 2025; Magnusson et al., 2025). This challenge is further exacerbated by the growing need to adapt LLMs to specialized downstream tasks under limited data or compute budgets (Thulke et al., 2024; Zhao et al., 2024b). As a result, a growing body of work studies how to automatically select a subset of examples from a large candidate pool that is most useful for a given target task—a problem commonly referred to as *targeted instruction selection* (Xia et al., 2024).

Despite increasing interest, the literature on targeted instruction selection remains fragmented and difficult to interpret (Appendix A). Proposed methods vary widely in their formulations, often involve multiple design choices (e.g., representations, similarity metrics, selection algorithms), and lack key baselines, such as the zero-shot baseline. Empirical results are inconsistent, and it remains unclear which techniques actually drive performance, when, and why.

In this work, we bring clarity to this space through a disentangled framework that separates the two core ingredients of targeted instruction selection: (i) the representation used to encode the data and (ii) the algorithm used to select examples based on these representations. This disentangled view allows us to isolate the effects of each component and enables controlled comparisons across models, datasets, and budgets. We further show that, despite differing in algorithmic detail, many selection methods can be unified under the view of approximate distance minimization between the selected subset and the target task distributions. We support this perspective with new generalization bounds that characterize when distance-based selection helps—and when it doesn't.

Our empirical findings are mixed but revealing. Gradient-based representations provide the most reliable distance-

*Equal contribution [1]Harvard University [2]MIT [3]Kempner Institute. Correspondence to: Nihal V. Nayak <nnayak@seas.harvard.edu>.

*Proceedings of the 43rd International Conference on Machine Learning*, Seoul, South Korea. PMLR 306, 2026. Copyright 2026 by the author(s).

to-query signal for query loss, but this signal does not consistently translate into downstream performance gains, and even the strongest methods fail to improve over zero-shot inference in some regimes (Section 5.1, Section 5.2, Appendix M, Appendix O). While no single method dominates, greedy round-robin selection tends to perform best at small budgets, while optimal transport-based methods offer modest gains at larger ones with Tulu V2 as the candidate pool (Section 5.2). However, our Dolci Instruct experiments show that these trends are candidate pool dependent and do not hold uniformly across tasks, models, and budgets (Appendix O). Surprisingly, randomly sampled subsets often match or outperform many popular selection methods (TyDiQA and MMLU-Pro in Figure 5), especially as the budget increases, highlighting the brittleness of current practice.

Our work makes the following contributions:

- We propose a **disentangled framework** for targeted instruction selection that isolates the effects of (i) data representations and (ii) selection algorithms, enabling controlled comparisons.
- We conduct **systematic empirical analyses** across multiple target tasks and LLMs, showing that gradient-based data representations provide the strongest distance-to-query signal for query loss, that greedy round-robin selection performs best at low budgets, and that optimal transport-based methods perform best at high budgets, using Tulu V2 as the candidate pool.
- We develop a **unified theoretical perspective** that interprets many selection algorithms as approximate distance minimization, and prove generalization bounds that explain both the benefits and limitations of this view.

## 2. Targeted Instruction Selection

Let $\mathcal{D} = \{z_i = (x_i, y_i)\}_{i=1}^N$ be a large candidate pool and $\mathcal{Q} = \{q_j = (x_j, y_j)\}_{j=1}^M$ be the query set drawn from the distribution $P_{\mathcal{T}}$ of the target task $\mathcal{T}$. We consider a model $f_\theta$ parameterized by $\theta$. For a sample $z = (x, y)$, we define the per-example loss as $\ell(\theta; z) := \ell(f_\theta(x), y)$. The goal of targeted instruction selection is to choose a subset of examples $\mathcal{S} \subseteq \mathcal{D}$ of size $B$ (training budget) and train the model $f_\theta$ using only examples in $\mathcal{S}$ so as to minimize the expected loss on the target task $\mathcal{T}$:

$$\mathcal{S}^* = \arg \min_{\substack{\mathcal{S} \subseteq \mathcal{D} \\ |\mathcal{S}|=B}} \mathbb{E}_{z \sim P_{\mathcal{T}}}[\ell(\theta_{\mathcal{S}}; z)].$$

Since we do not have access to the target tasks during training and selection, we use $\mathcal{Q}$ as a proxy for selection and choose $\mathcal{S}$ to solve the objective:

$$\hat{\mathcal{S}} = \arg \min_{\substack{\mathcal{S} \subseteq \mathcal{D} \\ |\mathcal{S}|=B}} \frac{1}{M} \sum_{j=1}^M \ell(\theta_{\mathcal{S}}; q_j).$$

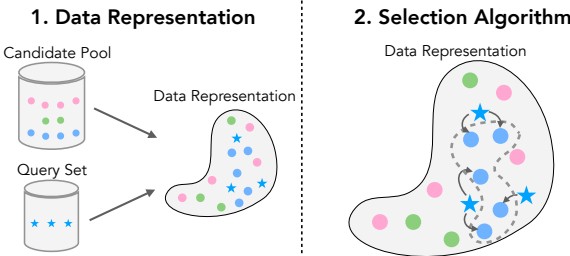

*Figure 1.* **Disentangled view of targeted instruction selection.** First, the query set (stars) and candidate pool (dots) are encoded as data representations. Then, for a given budget, using the data representations for the query and candidates, we perform targeted selection (denoted by the dotted line) using a selection algorithm such as greedy round-robin.

Optimizing this objective is impractical and combinatorially expensive, which has motivated compute efficient instruction selection methods.

## 3. A Disentangled View of Instruction Selection

In this section, we present the disentangled view of targeted instruction selection. Then, we describe the commonly used data representations and selection algorithms.

### 3.1. Disentangling Data Representation and Selection Algorithm

We adopt a disentangled view of instruction selection that separates two key components: (i) data representation, and (ii) the selection algorithm (Figure 1).

- **Data Representation**: First, we encode instruction-response pairs from the candidate pool and query sets into feature vectors (their *representations*). Ideally, these representations capture candidate-query distances that predict performance on the target task.
- **Selection Algorithm**: Next, the selection algorithm uses the data representations for the candidate pool and the query set to compute the similarity (or distances) between them and then select $B$ examples from the candidate pool based on the distances, where $B$ is the budget.

With this view, we aim to isolate the effects of data representations and selection algorithms from prior work that reports the best results on targeted instruction selection (Liu et al., 2024b; Xia et al., 2024; Ivison et al., 2025).

### 3.2. Data Representation

Here, we describe three data representation approaches for encoding the samples (More details in Appendix E.2). In this work, for a fair comparison, all these data representations are used to compute a cosine similarity matrix between the query and the candidate samples.

**RDS+.** We refer to the data representation from Ivison et al. (2025) as RDS+ and discuss the selection algorithm (RR) in Section 3.3. RDS+ computes the representations for the samples in the query and candidate pool by taking a position-weighted mean of the hidden states from the base language model (Muennighoff, 2022).

**EMBED.** Here, the query and candidate samples are passed through an off-the-shelf sentence encoder, such as GTR-T5 (Ni et al., 2022), to produce representations. Since these are typically much smaller, they significantly reduce FLOPs compared to RDS+. We follow the setup from Ivison et al. (2025) to produce these representations.

**LESS.** Xia et al. (2024) propose LESS (Low-rank gradiEnt Similarity Search), an optimization-aware influence formulation that represents query and candidate samples as low-dimensional gradient features. LESS estimates influence using a first-order approximation of training dynamics (Pruthi et al., 2020) and adapts it to Adam (Kingma & Ba, 2015). Concretely, it averages the cosine similarity between the query gradient and the candidate's Adam update vector across multiple training checkpoints. To make this approach scalable, LESS computes LoRA gradients (Hu et al., 2022) and applies random projection to obtain compact vectors, resulting in an efficient and reusable gradient datastore (Johnson & Lindenstrauss, 1984; Park et al., 2023). LESS is part of a broad literature on data attribution and influence estimation, which seeks to quantify the effect of individual training examples on a model (Grosse et al., 2023; Kwon et al., 2024; Ruis et al., 2025; Chang et al., 2025; Wang et al., 2025). For simplicity, we use LESS representations to encode the candidate pool and query sets (Appendix D for details).

### 3.3. Selection Algorithm

We describe selection algorithms for targeted instruction selection (More details in Appendix F).

**Greedy Round-Robin (RR).** We consider the greedy round-robin algorithm from Ivison et al. (2025). For each query sample, RR selects the sample from the candidate pool with the highest cosine similarity, adds the sample to the subset, and then removes it from the candidate pool. The algorithm repeats the round-robin process across all query samples until the data selection budget $B$ is exhausted.

**Doubly Greedy (DG).** We consider the doubly greedy selection algorithm from Xia et al. (2024). Given a similarity matrix, DG assigns each candidate sample an influence score equal to the maximum similarity it has with any query point. Then, DG selects the top-$B$ influential candidates to create the subset.

**KNN-Uniform.** KNN-Uniform is a selection algorithm inspired by optimal transport (Liu et al., 2024b). KNN-Uniform proposes a closed-form solution based on $K$ nearest neighbors to avoid explicitly solving the optimal transport problem. The algorithm first determines $K$ based on the trade-off between the alignment and diversity. After determining $K$, for each query, a uniform probability mass is assigned to the $K$ nearest neighbors, and then top-$B$ candidates with the highest mass (summed over queries) are selected.

**KNN-KDE.** KNN-KDE builds on the closed-form solution of KNN-Uniform by adding an additional regularizer to reduce the effect of near duplicates in the candidate pool (Liu et al., 2024b). Instead of assigning uniform mass to nearly identical points, Liu et al. (2024b) proposes incorporating kernel density estimation (KDE) as a regularizer. Then, for each query example, the $K$ nearest neighbors are assigned probability mass weighted by the inverse of their density estimates, and the top-$B$ candidates are selected.

**Unbalanced OT (UOT).** Here, we propose a new selection algorithm based on unbalanced optimal transport (Chizat et al., 2018), which we refer to as UOT. Unlike KNN-Uniform, in UOT, we explicitly solve the optimization problem of transporting mass from the query set to the candidate pool, penalizing marginal deviations (Appendix C). This allows us to "ignore" outliers and less relevant samples for the target task. After solving for the transport plan, we sum the plan over the rows (queries) and choose the top-$B$ candidates with the highest mass (Appendix F.2 for implementation).

## 4. Experimental Setup

We closely follow the setup from Ivison et al. (2025) and Xia et al. (2024). Following Xia et al. (2024), we primarily use Llama-2-7B as the base model and train on the selected subset of training data. In Appendix M, we include experiments with additional models. Across all data representations, selection algorithms, and models, we use cosine similarity (or cosine distance) to measure the similarity between the query-candidate pairs. We experiment with the following target tasks: BBH, Codex, GSM8K, TyDiQA, and MMLU-Pro (Appendix B). We use the subsampled Tulu V2 dataset from Ivison et al. (2025) as the candidate pool. More training details and hyperparameters are included in Appendix G. We use the evaluation code from Ivison et al. (2025) and `lm-eval-harness` to evaluate the models and report the downstream metrics. We also report the average cross entropy loss over the response tokens on the query set (query loss).

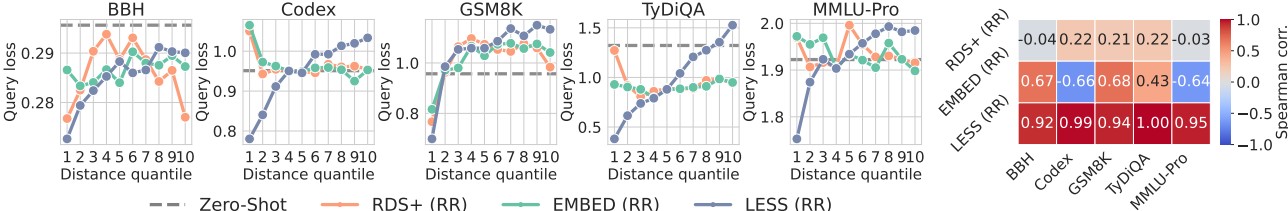

*Figure 2.* **Query loss vs. subset-query distance quantile.** We stratify candidates into 10 distance quantiles (1 = closest, 10 = farthest) using each representation, select 500 examples per quantile using the RR selection algorithm, and train the Llama 2 7B model. We report query-set cross-entropy loss and Spearman correlation per target task. LESS (RR) exhibits a strong monotonic increase in loss with distance (high positive Spearman correlation), whereas RDS+ (RR) and EMBED (RR) show weak or inconsistent correlations.

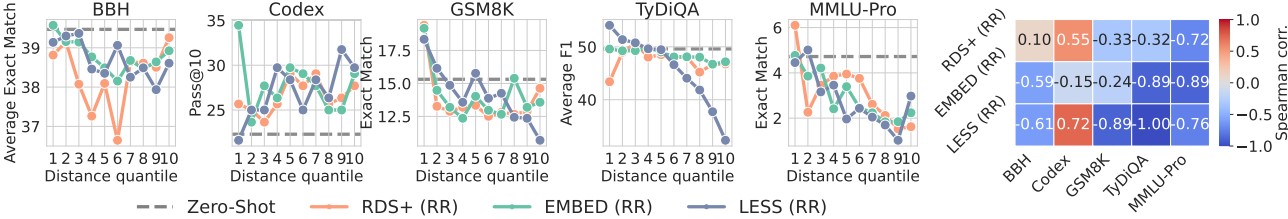

*Figure 3.* **Downstream performance vs. subset-query distance quantile.** Using the same quantile construction and training protocol as Figure 2, we evaluate downstream task performance across distance quantiles and report Spearman correlation per target task. LESS (RR) shows a strong negative correlation across most target tasks, while RDS+ (RR) and EMBED (RR) exhibit weaker, less consistent trends.

## 5. Experiments and Analysis

### 5.1. Does Subset Distance to Query Predict Performance?

In this experiment, we aim to understand the relationship between the subset distance to the query and performance. The goal is to determine which data representation creates subsets whose distances predictably correlate with loss and downstream performance. If there is a correlation, we can formulate optimization objectives to minimize the distance between the query set and the candidates, thereby achieving the highest downstream performance. In Section 6, we also theoretically motivate this view by showing that minimizing the subset-query distance translates into minimizing the loss.

**Setup.** We consider all the data representation methods from Section 3.2. Given a cosine similarity matrix, we run the greedy round-robin selection algorithm over the candidates to obtain an ordering, which we then use to stratify them into distance quantiles. The quantiles are indexed from 1 to 10, where 1 corresponds to the subset closest to the query and 10 to the farthest. Then, we select the top 500 training examples from each quantile and train Llama 2 7B on them.

**Results.** Figure 2 shows that only quantiles created by LESS (RR) highly correlate with the loss on the query set. On the other hand, both RDS+ (RR) and EMBED (RR) show very low Spearman correlation. Across all target tasks,

models trained on a subset in the first distance quantile using LESS (RR) achieve the lowest loss. In contrast, RDS+ and EMBED may not often show the lowest loss in their first quantile. To our surprise, we find that both methods can sometimes have the lowest loss in the last distance quantile (see BBH for RDS+ (RR) and MMLU-Pro for EMBED (RR)). Figure 3 shows that LESS (RR) shows a strong negative Spearman correlation (lower is better) across four out of the five downstream tasks. However, we do find that lower loss in Figure 2 does not necessarily translate to the highest downstream performance in the first quantile. For instance, we see that EMBED (RR) performs better than LESS (RR) on four out of the five target tasks despite having a much higher loss on the query set. This discrepancy motivates the need for data representations that not only correlate with the loss but also translate to better downstream performance. Finally, we analyze the behavior of these representations by further subdividing the first quantile in Appendix H.

Overall, LESS is the only representation that creates subsets whose distance correlates predictably with query loss and downstream performance. In Appendix M, across additional models, we find that LESS consistently shows strong correlation across target tasks, but the trends are less consistent on downstream performance with newer, over-trained models. Finally, in Appendix O, when we used Dolci Instruct as the candidate pool, we again see that LESS strongly correlates with query loss but is less consistent in downstream performance.

**Loss on the Query Set across Subset Budgets for Different Data Representations (Llama 2 7B)**

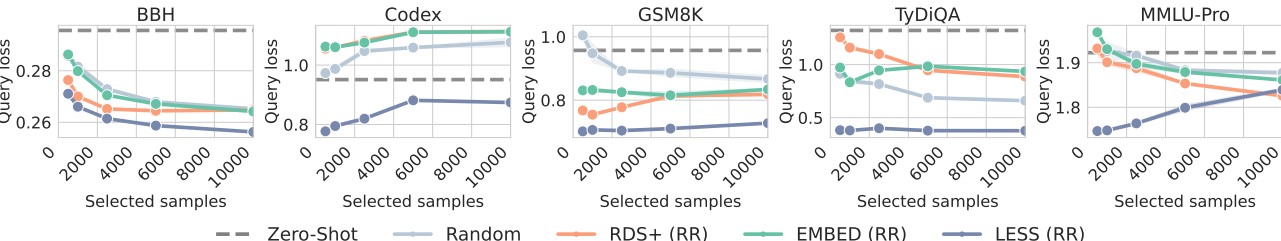

*Figure 4.* **Query loss vs. budget for different data representations (fixed selection algorithm).** Using greedy round-robin selection and the query-candidate pool similarity, we select subsets of size $B \in \{500, 1000, 2500, 5000, 10000\}$, train Llama 2 7B on them, and report average cross entropy loss averaged across three seeds and the standard error. Random averages over three uniformly sampled subsets from the candidate pool. LESS (RR) achieves the lowest loss across target tasks, while RDS+ (RR) and EMBED (RR) can underperform Random at larger budgets.

**Performance on the Test Set across Subset Budgets for Different Data Representations (Llama 2 7B)**

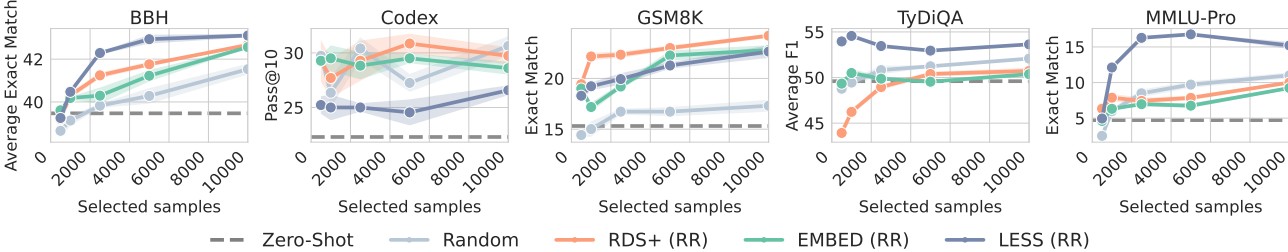

*Figure 5.* **Downstream performance vs. budget for different data representations (fixed selection algorithm).** With the same greedy round-robin selection and budgets as Figure 4, we report the downstream performance for different data representations averaged across three random seeds and the standard error. LESS (RR) performs best on BBH, TyDiQA, and MMLU-Pro, whereas RDS+ (RR) performs the best on GSM8K and is competitive with Random on Codex.

## 5.2. Effect of Data Representation across Subset Budgets

Here, we select subsets with different data representations while keeping the selection algorithm fixed to study performance as the budget increases.

**Setup.** We fixed the selection algorithm to a greedy round-robin approach, since Ivison et al. (2025) showed strong performance across tasks. Given a cosine similarity matrix from a data representation method, we select B samples from the candidate pool where $B \in \{500, 1,000, 2,500, 5,000, 10,000\}$. Then, we train the base model, Llama 2 7B, on the selected samples and report the downstream performance. We also include a Random baseline that uniformly samples $B$ candidates without replacement from the candidate pool. For all methods, we train three models with different seeds and report the average performance and standard error. For the Random baseline, we sample three times from the candidate pool, train one model on each, and report the average performance.

**Results.** Figure 4 shows that LESS (RR) achieves the lowest loss on the query set across all the target tasks under different budget constraints. We observe that RDS+ (RR) and EMBED (RR) can have higher loss than the Random

baseline. Finally, we see that the query loss can increase or may stabilize as we continue increasing the budget. Figure 5 shows that LESS (RR) outperforms the other methods by a clear margin on three out of the five target tasks. Next, we observe that RDS+ (RR) achieves higher performance across all budget constraints on GSM8K, but Random is a competitive baseline at larger budgets on Codex. Lastly, we note that Random is a strong baseline but often underperforms targeted instruction selection methods at low budgets.

Overall, these results show that, under a fixed selection algorithm (greedy round-robin), gradient-based representations (LESS) perform better than model-based representations on a majority of the target tasks. Across models and downstream target tasks that benefit from additional training (i.e., zero-shot performance is not saturated), we see that LESS often outperforms other baselines (Appendix M). When using Dolci Instruct as the candidate pool (Appendix O), RDS+ and EMBED either match or outperform LESS on downstream metrics, suggesting that the performance of the data representation is candidate pool dependent. A key limitation of LESS representations is that they are computationally more expensive than model-based representations, as they require a forward and a backward pass over the entire candidate pool. For this reason, in

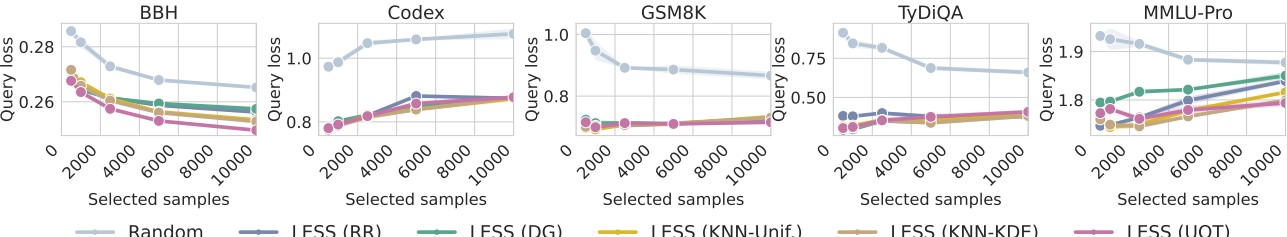

*Figure 6.* **Query loss vs. budget for different selection algorithms (fixed data representation).** Using LESS representations and the query-candidate pool cosine similarity (or distance), we select subsets of size $B \in \{500, 1000, 2500, 5000, 10000\}$ with each selection algorithm, train Llama 2 7B on them, and report average cross entropy loss on the query set averaged across three seeds and the standard error. Random averages over three uniformly sampled subsets from the candidate pool. UOT achieves the lowest loss on three of the five datasets and remains competitive on the others, while DG often underperforms, yielding the highest loss on three datasets.

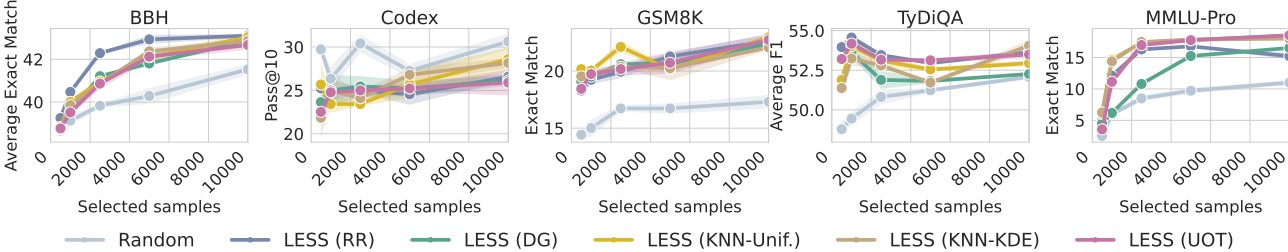

*Figure 7.* **Downstream performance vs. budget for different selection algorithms (fixed data representation).** With the same data representation and the budgets as Figure 6, we report downstream performance for different selection algorithms averaged across three seeds and the standard error. RR tends to perform best at smaller budgets, whereas UOT and KNN-KDE perform better at larger budgets; DG consistently underperforms across three of the five datasets.

Appendix J, we further investigate whether subsets created by smaller proxy LESS models offer a cheaper alternative, and show that even models with 135M parameters can select examples for training larger models and dramatically reduce FLOPs. Finally, we find that subsets selected by model-based representations are competitive and often outperform Random under constrained budgets.

### 5.3. Effect of Selection Algorithms across Subset Budgets

Here, we select subsets with different selection algorithms while keeping the data representation fixed to study the performance as the budget increases.

**Setup.** The experimental setup closely follows Section 5.2. Building on our insights from Section 5.2, we fix LESS as our data representation. Given the LESS representations for the query set and candidate pool, we use the selection algorithm to select samples for budgets $B \in \{500, 1,000, 2,500, 5,000, 10,000\}$. We use the weighted cosine similarity matrix between the query and the candidate pool (Equation 2 in the Appendix) to select instructions using greedy round-robin (RR) and doubly greedy (DG). We convert this cosine similarity matrix into a cosine dis-

tance matrix to select samples with KNN-Uniform, KNN-KDE, and UOT (Appendix F.1 and F.2). Following Liu et al. (2024b), we also include results for KNN-Uniform and KNN-KDE using L2 distance in Appendix K.

**Results.** Figure 6 shows that across the five tasks, UOT achieves the lowest loss on three of them, while remaining competitive on the others. On the other hand, we see that DG often has the highest loss across tasks at large budgets. Figure 7 shows the downstream performance after training Llama-2-7B on the selected subsets. We observe that LESS (RR), the greedy round-robin selection algorithm, performs well on low budgets. On the other hand, LESS (UOT) and LESS (KNN-KDE) perform better with higher budgets. Finally, we find that the doubly greedy (DG) algorithm consistently underperforms other selection algorithms, suggesting that it might be choosing samples closest to only a subset of the queries.

Overall, our results show that greedy round-robin (RR) selection works best under limited budget constraints, but these gains generally diminish as more examples are added. At larger budgets, optimal transport–based selection methods (UOT and KNN-KDE) provide greater benefits. In Appendix M, we find that the performance

trends observed in this experiment generalize to other models. For example, LESS (RR) performs best on BBH with Llama 3.2 3B and Qwen3 4B Base. We also observe that LESS (UOT) performs best on MMLU-Pro at high budgets with Olmo 3 7B Base. These results suggest that selection algorithms exhibit similar performance trends across models for particular target tasks. However, when we change the candidate pool to Dolci Instruct (Appendix O), we see that no selection algorithms consistently show strong budget-dependent trends, which suggests the effect of selection algorithms are dependent on the candidate pool.

## 6. A Unifying View: Instruction Selection as Set Distance Minimization

While instruction selection algorithms vary in form, from greedy heuristics to transport-based selection, they often share a deeper, unifying principle: selecting a subset that is "close" to the query set in the representation space. In this section, we formalize and motivate this perspective.

Concretely, we observe that a broad class of existing methods (including those discussed in Section 3.3) can be viewed as approximately minimizing a distributional distance between the selected subset $\mathcal{S}$ and the query set $\mathcal{Q}$:

- Greedy algorithms (e.g., round-robin, doubly greedy) aim to reduce the average or maximum similarity-based distance between $\mathcal{S}$ and $\mathcal{Q}$.
- Density-based methods (e.g., KNN-Uniform, KNN-KDE) select points from high-density regions near $\mathcal{Q}$, implicitly matching the mass of the subset distribution to the query.
- Optimal transport approaches (e.g., UOT) directly solve for a transport coupling minimizing cost between empirical distributions associated with $\mathcal{S}$ and $\mathcal{Q}$.

Despite differing in implementation and exact objective, all these methods can be understood as minimizing a distance function $\mathrm{Dist}(\mathcal{S}, \mathcal{Q})$, where the choice of distance reflects the algorithm's inductive biases. This unified view not only clarifies relationships between methods, but also explains their performance patterns. In the remainder of this section:

- We show that reducing the subset-query distance tightens a generalization bound on downstream loss (Section 6.1).
- We show that the benefit of distance-aware selection over random sampling diminishes with budget, and characterize this tradeoff (Section 6.2).

### 6.1. Minimizing Subset-Query Distance Tightens a Generalization Bound

We establish a theoretical bound formalizing the intuition that selecting a subset $\mathcal{S}$ whose empirical distribution is close to that of the query set $\mathcal{Q}$ leads to improved performance on the target task $\mathcal{T}$. We show that the empirical

test loss $L_{\mathcal{T}}(\theta_S) := \frac{1}{|\mathcal{T}|} \sum_{z \in \mathcal{T}} \ell(\theta_S; z)$ for a loss function $\ell(\theta_S; z)$ and $\theta_{\mathcal{S}} \in \arg\min_{\theta \in \Theta} L_{\mathcal{S}}(\theta)$ an empirical risk minimizer trained on $\mathcal{S}$, is upper-bounded by the 1-Wasserstein distance between the empirical distributions of the selected subset and the query set, denoted by $W_1(\hat{P}_{\mathcal{S}}, \hat{P}_{\mathcal{Q}})$. Here, $\hat{P}_{\mathcal{S}}$ and $\hat{P}_{\mathcal{Q}}$ denote the empirical distributions associated with $\mathcal{S}$ and $\mathcal{Q}$, respectively. This result follows by viewing subset selection as a two-stage domain adaptation problem: first from $\mathcal{S}$ (source) to $\mathcal{Q}$ (target), and then from $\mathcal{Q}$ (source), to $\mathcal{T}$ (target). Bounds of this form are well established in the domain adaptation literature, where optimal transport distances between source and target distributions are known to control transfer error (Redko et al., 2017; Courty et al., 2017). In our setting, this bound implies that algorithms which approximately minimize $W_1(\hat{P}_{\mathcal{S}}, \hat{P}_{\mathcal{Q}})$ directly tighten a theoretical upper bound on target task performance.

**Theorem 6.1.** *Let $\ell : \Theta \times \mathcal{Z} \to \mathbb{R}_+$ be a loss function, where $\mathcal{Z} \subset \mathbb{R}^d$ denotes the data space and $\Theta$ the parameter space. Assume that $\ell$ is symmetric, convex, bounded, satisfies the triangle inequality, and for $z = (x, y) \in \mathcal{Z}$ admits the parametric form $\ell(\theta; z) = |y - f_\theta(x)|^q$ for some $q > 0$. Let $\mathcal{D}$ denote a labeled candidate pool, and let $\mathcal{S} \subseteq \mathcal{D}$ be any subset of size $B := |\mathcal{S}|$. Let $\mathcal{Q}$ (the query set) and $\mathcal{T}$ (the test set) be labeled datasets, and assume $|Q| \leq \min(|\mathcal{T}|, |\mathcal{S}|)$. Then for any $d' > d$ and $c' < \sqrt{2}$, there exists a constant $N_0$, depending on $d'$, such that for any $\delta > 0$ and $|Q| \geq N_0 \max(\delta^{-(d'+2)}, 1)$, with probability at least $1 - 2\delta$:*

$$L_{\mathcal{T}}(\theta_{\mathcal{S}}) \leq \underbrace{W_1(\hat{P}_{\mathcal{S}}, \hat{P}_{\mathcal{Q}})}_{\substack{\text{Subset and query} \\ \text{dataset distance}}} + \underbrace{W_1(\hat{P}_{\mathcal{Q}}, \hat{P}_{\mathcal{T}})}_{\substack{\text{Query and test} \\ \text{dataset distance}}}$$
$$+ \underbrace{L_{\mathcal{S}}(\theta_{\mathcal{S}})}_{\text{training error}} + \zeta \sqrt{\frac{2}{c'} \log\left(\frac{1}{\delta}\right)} + \tilde{\lambda} \quad (1)$$

*where $W_1$ is the 1-Wasserstein distance, $\zeta$ is a constant given by $\zeta := B^{-\frac{1}{2}} + 2|\mathcal{Q}|^{-\frac{1}{2}} + |\mathcal{T}|^{-\frac{1}{2}}$, and $\tilde{\lambda}$ is the minimum combined error $L_{\mathcal{S}}(\theta_{\tilde{\mathcal{S}}}) + 2L_{\mathcal{Q}}(\theta_{\tilde{\mathcal{S}}}) + L_{\mathcal{T}}(\theta_{\tilde{\mathcal{S}}})$ over datasets $\tilde{\mathcal{S}} \subseteq \mathcal{D}$ of size $B$.*

*Proof in Appendix L.2*

Among the terms in equation 1, only $W_1(\hat{P}_{\mathcal{S}}, \hat{P}_{\mathcal{Q}})$ is directly affected by the choice of $\mathcal{S}$ (for fixed $\mathcal{Q}, \mathcal{T}$). Consequently, selecting $\mathcal{S}$ to (approximately) minimize $W_1(\hat{P}_{\mathcal{S}}, \hat{P}_{\mathcal{Q}})$ is a principled objective: any reduction in this transport distance directly tightens the right-hand side, and hence the worst-case upper bound on $L_{\mathcal{T}}(\theta_{\mathcal{S}})$. The second Wasserstein term, $W_1(\hat{P}_{\mathcal{Q}}, \hat{P}_{\mathcal{T}})$, quantifies mismatch between the query distribution and the downstream evaluation distribution. This term is independent of $\mathcal{S}$, and therefore sets an irreducible error on how informative $\mathcal{Q}$ is about $\mathcal{T}$.

At the same time, the theorem does not imply that minimizing $W_1(\hat{P}_{\mathcal{S}}, \hat{P}_{\mathcal{Q}})$ alone guarantees strong target

performance. The bound also includes the training error $L_{\mathcal{S}}(\theta_{\mathcal{S}})$ and the term $\tilde{\lambda}$. In the instruction-selection setting, these terms capture two additional failure modes: (i) $\mathcal{S}$ may be close to $\mathcal{Q}$ yet noisy or internally inconsistent, leading to large $L_{\mathcal{S}}(\theta_{\mathcal{S}})$; and (ii) there may be no single hypothesis that performs well simultaneously on $\mathcal{S}$, $\mathcal{Q}$, and $\mathcal{T}$, leading to large $\tilde{\lambda}$. The quantity $\tilde{\lambda}$ can be interpreted as an ideal joint error term: it is small when the candidate pool $\mathcal{D}$ contains a size-$B$ subset whose ERM achieves low loss on $\mathcal{S}$, $\mathcal{Q}$, and $\mathcal{T}$ simultaneously. Consequently, if $\mathcal{D}$ lacks coverage of the skills required by $\mathcal{Q}$ and $\mathcal{T}$, or if the hypothesis class cannot realize a predictor that works well across these datasets, then no selection rule based purely on distribution matching can guarantee strong downstream performance. The remaining terms account for finite-sample effects through $\zeta$.

### 6.2. Diminishing Returns of Query-Aware Selection as Budget Increases

Computing the exact subset that minimizes the distance to the query set is a combinatorial problem, and even approximate distance-minimization methods can be computationally expensive. In contrast, uniformly sampling $B$ samples from the candidate pool $\mathcal{D}$ is essentially cost-free and becomes increasingly competitive as the selection budget $B$ grows. For analysis, we consider the random baseline obtained by sampling $B$ points i.i.d. from the empirical pool distribution $\hat{P}_{\mathcal{D}}$ (i.e., with replacement). Then, for a fixed budget $B$, Theorem 6.2 formalizes the benefit of query-aware selection by upper bounding the improvement in test loss achieved by training on a Wasserstein-optimal subset—i.e., the subset closest to $\mathcal{Q}$—relative to this random sample of the same size.

**Theorem 6.2.** *Let the candidate pool $\mathcal{D} \subset \mathbb{R}^d$ lie in a set of diameter $\Delta$ and $d \geq 3$. Let $\mathcal{S}^{\mathrm{rnd}} := (s_1, \ldots, s_B)$ be a tuple of $B$ elements sampled i.i.d. uniformly from $\mathcal{D}$, and let $S_W^* \subseteq \mathcal{D}$ be a subset of size $B$ that minimizes the 1-Wasserstein distance to the query set $\mathcal{Q}$, i.e., $S_W^* \in \arg\min_{\mathcal{S} \subseteq \mathcal{D}; |\mathcal{S}|=B} W_1(\hat{P}_{\mathcal{S}}, \hat{P}_{\mathcal{Q}})$. Assume (i) $L_{\mathcal{S}}(\theta)$ is $\mu$–strongly convex in $\theta$, (ii) $L_{\mathcal{T}}(\theta)$ is $K$-Lipschitz, and (iii) $\nabla_{\theta}\ell(\theta; z)$ is $G_{\theta z}$-Lipschitz with respect to $z$. Then, there exists a constant $C_d > 0$, depending only on the dimension $d$, such that, with probability at least $1 - 2\delta$:*

$$L_{\mathcal{T}}(\theta_{\mathcal{S}^{\mathrm{rnd}}}) - L_{\mathcal{T}}(\theta_{S_W^*})$$

$$\leq C_\star \left( \underbrace{C_d \Delta B^{-1/d}}_{\substack{\text{Curse of} \\ \text{dimensionality}}} + \underbrace{\Delta \sqrt{\frac{\log(1/\delta)}{2B}}}_{\text{Concentration bound}} \right.$$

$$\left. + \underbrace{W_1(\hat{P}_{\mathcal{D}}, \hat{P}_{\mathcal{Q}})}_{\text{Pool-query mismatch}} + \underbrace{W_1(\hat{P}_{S_W^*}, \hat{P}_{\mathcal{Q}})}_{\text{Distance residual}} \right)$$

*Proof in Appendix L.3.*

Theorem 6.2 shows that the upper bound on the potential

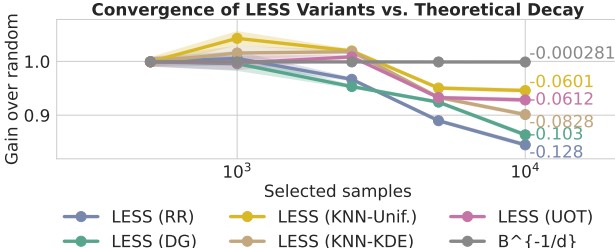

*Figure 8.* **Convergence of LESS variants toward random sampling as budget increases.** We plot the loss gap between random sampling and the LESS variants as a function of budget $B$ (log-scale) on MMLU-Pro. The gray reference line indicates the $B^{-1/d}$ decay predicted by Theorem 6.2. We report the average difference in loss across one seed of LESS and three randomly sampled multi-sets of size $B$, along with the standard error. For visual comparison of decay rates, we apply a constant offset to each LESS curve so that all methods start at $B_0^{-1/d}$ where $B_0 = 500$ and $d = 8192$.

improvement in test loss from training on a query-aware subset $S_W^*$ decreases as the budget $B$ increases. In particular, when $d \geq 3$, the leading $B$-dependent term in the upper bound scales as $C_\star C_d \Delta B^{-1/d}$ (the $B^{-1/2}$ concentration term is lower order). Thus, up to the additive shift induced by pool–query mismatch $W_1(\hat{P}_{\mathcal{D}}, \hat{P}_{\mathcal{Q}})$ (and the residual $W_1(\hat{P}_{S_W^*}, \hat{P}_{\mathcal{Q}})$), we can define a critical subset size for a tolerance $\varepsilon$: requiring $C_\star C_d \Delta B^{-1/d} \leq \varepsilon$ yields $B \geq (C_\star C_d \Delta / \varepsilon)^d$. This highlights the curse of dimensionality: in high-dimensional embedding spaces, making random sampling competitive can require $B$ to grow on the order of $\varepsilon^{-d}$, whereas query-aware selection can be especially beneficial at substantially smaller budgets. The intrinsic dimension of these features could be much lower, which would further tighten the bound (Weed & Bach, 2017).

Figure 8 shows the change in performance gap between LESS variants and random sampling as the budget increases on MMLU-Pro. As $B$ increases, the gap shrinks, and the performance approaches the random sampling, which is qualitatively aligned with Theorem 6.2. While $B^{-1/d}$ reference rate shows a slow worst-case decay in high dimensions, we see that LESS (KNN-Unif.), LESS (KNN-KDE), and LESS (UOT) exhibit a slower or a similar decay up to 2,500 samples, suggesting the bound is non-trivial in this regime. Finally, the LESS variants approach the random baseline at different rates, suggesting that the choice selection algorithm affects the constants and residual errors.

## 7. Related Work

**Subset Selection.** Subset selection (or coreset selection) is a fundamental task in machine learning, where the goal is to choose a subset of the data from a large candidate pool for training and achieve downstream performance similar to or better than that of training on the entire candidate pool (Wei et al., 2015; Huang et al., 2019; Moser

et al., 2025). Over the years, several techniques involving clustering (Har-Peled & Mazumdar, 2004; Chen et al., 2023), gradient matching (Killamsetty et al., 2021), kernel thinning (Dwivedi & Mackey, 2024; Carrell et al., 2025), and proxy models (Ye et al., 2025; Magnusson et al., 2025) have been proposed to efficiently select core training samples in a variety of domains (Hulkund et al., 2025). In this work, we focus on selecting a subset of instruction-response pairs from a candidate pool for a target task.

**Instruction Selection.** Instruction selection is a key ingredient in today's large language model training pipeline (Olmo et al., 2025). While curation via careful experimentation in the post-training pipeline has led to dramatic performance improvements (Longpre et al., 2023; Guha et al., 2025), there is a growing interest in automatically curating instruction tuning datasets (Liu et al., 2024a; Yin & Rush, 2025). We focus on instruction selection for target tasks, selecting samples within a given budget from a candidate pool. Existing work on automatic instruction selection often relies on heuristics such as length (Zhao et al., 2024a), perplexity (Ankner et al., 2024), number of reasoning (Li et al., 2025), task similarity (Xia et al., 2024; Ivison et al., 2025; Nikdan et al., 2025), and more (Albalak et al., 2024). Our focus is to understand targeted instruction selection that uses similarity between the query and the candidate pool to select subsets.

## 8. Conclusion

We disentangle the key components of targeted instruction selection and provide new critical empirical and theoretical insights. Our experiments reveal that gradient-based data representations create subsets whose distances to the query strongly predict performance. Exploring computationally cheaper alternatives to produce such representations is a promising research direction (Appendix J). Our theoretical insight, which unifies several existing selection algorithms as approximate distance minimizers, provides a general framework for creating new selection algorithms. Overall, we present a practical roadmap for designing improved targeted instruction selection frameworks.

## Acknowledgments

We thank Yonatan Belinkov, Bingbin Liu, Lyndon Lam, and the members of the ML Foundations group and the Kempner Institute for thoughtful feedback on the manuscript. Nihal V. Nayak, Paula Rodriguez-Diaz, and David Alvarez-Melis acknowledge support from the NSF AI-SDM Institute (Award No. 2229881), the Kempner Institute, the FAS Dean's Competitive Fund for Promising Scholarship, and the Aramont Fellowship Fund. Neha Hulkund is supported by the NSF GRFP DGE-2146755. Sara Beery acknowledges support by the Schmidt Sciences AI2050 Program, NSF Awards No. 2330423 and 2441060, NSERC Award No. 585136, and the MIT-IBM Watson AI Lab.

## Impact Statement

This paper presents work aimed at advancing targeted instruction selection in large language models (LLMs). While our work comprehensively evaluates numerous LLMs across many target tasks, we rely on existing pre-trained LLMs. Further fine-tuning these pre-trained LLMs might amplify any pre-existing biases or might even hurt performance on unrelated tasks. Before deploying any of these models, we recommend conducting careful evaluations on target tasks and additional safety checks.

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

| Existing Work | Data Representation | Similarity | Selection Algorithm | Candidate Pool | Budget |
|---|---|---|---|---|---|
| Xia et al. (2024) | LESS | Cosine sim. | Doubly Greedy (DG) | Tulu V1 (Subset) | 13,533 |
| Liu et al. (2024b) | LESS | L2 dist. | KNN-KDE, KNN-Unif. | Tulu V1 (Subset) | Multiple |
| Ivison et al. (2025) | RDS+ | Cosine sim. | Round-Robin (RR) | Tulu V2 | 10,000 |

*Table 1.* **Overview of the fragmented literature on targeted instruction selection.** We summarize prior work and highlight differences in data representation, similarity between data representations, selection algorithm, candidate pool, and selection budget, which prevent us from systematically understanding which of these key factors contributes to performance. Note that Xia et al. (2024) uses 5% of the candidate pool as the budget, resulting in 13,533 samples whereas Liu et al. (2024b) uses 0.5%, 1.0%, and 5.0% of the candidate pool as the budget.

## A. Fragmented Literature on Targeted Instruction Selection

Table 1 gives an overview of the fragmented literature on targeted instruction selection. We show that existing work, notably Xia et al. (2024), Liu et al. (2024b), and Ivison et al. (2025), widely vary in their data representations, similarity metrics, selection algorithm, candidate pool, and budget. While Xia et al. (2024) and Liu et al. (2024b) are similar in many ways, we observed that the selection algorithm in Liu et al. (2024b) makes additional changes to the data representation construction process and uses L2 distances between query and the candidates (Appendix K). These small but key differences prevent us from systematically comparing the selection algorithms, KNN-KDE and KNN-Uniform, introduced in Liu et al. (2024b). Finally, Ivison et al. (2025) uses Tulu V2 mixture (Ivison et al., 2023) as the candidate pool but also changes the data representation, similarity metric, and budget. For these reasons, we aim to bring clarity to this important but rather fragmented literature through systematic experiments.

## B. Target Tasks

We closely follow the target tasks from Ivison et al. (2025). In addition, we include the MMLU-Pro dataset (Wang et al., 2024). Table 2 provides statistics of the target tasks used in this work. Below, we provide descriptions of the target tasks:

**BBH (BIG-Bench Hard).** BBH is a curated subset of tasks from BIG-Bench that are empirically difficult for language models without advanced reasoning strategies. The benchmark includes tasks such as logical deduction, causal reasoning, temporal reasoning, symbolic manipulation, and algorithmic problem solving. BBH is commonly used to assess emergent reasoning abilities and sensitivity to prompting strategies, particularly few-shot and chain-of-thought prompting. We use the few-shot examples across all subtasks ($3 \times 27 = 81$ samples) as our query set. In the query set, all the few-shot examples are treated as individual examples. We follow the evaluation in (Suzgun et al., 2023) and evaluate the model with a 3-shot chain-of-thought. We report the average exact match on the test set.

**Codex.** Codex is an evaluation suite for coding-based models based on the correctness of completing coding tasks at scale, testing advanced reasoning ability over short and long-context coding tasks (Chen et al., 2021). We closely follow the custom query-set/test-set split from Ivison et al. (2025), which divides the existing dataset of 164 examples into a query set of 16 samples and uses the remaining examples as the test set. For downstream evaluation on the test set, we report the pass@10 with a sampling temperature of 0.8.

**GSM8K.** GSM8K is a dataset of grade-school-level math word problems that require multi-step arithmetic reasoning. Each problem is paired with a detailed, step-by-step solution that explicitly outlines the reasoning process leading to the final answer. The dataset is widely used to evaluate numerical reasoning, chain-of-thought capabilities, and the ability of language models to perform symbolic manipulation and logical decomposition rather than surface-level pattern matching. We treat the 8-shot examples as individual samples and use them as our query set. We evaluate the test set with the 8 in-context samples using the chain-of-thought and report the exact match.

**TyDiQA.** TyDiQA is a multilingual question answering dataset built from Wikipedia that contains real information-seeking questions written by native speakers in 11 typologically diverse languages, avoiding translation artifacts. The dataset is widely used to evaluate multilingual and cross-lingual QA robustness, especially under linguistic diversity and distribution shift. We follow Ivison et al. (2025) and evaluate the models in a 1-shot setting across 9 languages where the answer is provided in the passage. We use the 1-shot samples across the languages as the query set.

| Target Task | Num. Query Set Samples | Num. Test Set Samples | Target Task Evaluation |
|---|---|---|---|
| BBH (BIG-Bench Hard) | 81 | 6,511 | 3-Shot |
| Codex | 16 | 148 | 0-Shot |
| GSM8K | 8 | 1,319 | 8-Shot |
| TyDiQA | 9 | 5,077 | 1-Shot |
| MMLU Pro | 70 | 12,032 | 0-Shot |

*Table 2.* **Statistics of target tasks used in our experiments.** The query set is used to select samples from the candidate pool to train the base model, and then evaluated on the test set from the target task. Each target task is evaluated using 0 to N few-shot in-context examples, and we reuse the query set samples as these few-shot examples if mentioned in Appendix B.

**MMLU Pro.** MMLU-Pro is designed to evaluate advanced reasoning across a broad set of academic and professional domains, emphasizing multi-step reasoning and conceptual understanding, using carefully curated questions with reduced surface cues and strong controls against memorization or data leakage. It is commonly used to probe the upper limits of model reasoning performance in settings that more closely reflect expert-level problem solving. We treat the few-shot samples in the validation set as individual samples and use them as our query set. We perform zero-shot chain-of-thought evaluation on the test set and report exact match scores.

In contrast to Ivison et al. (2025), we do not include MMLU and SQuAD in the main experiments, as these datasets are largely saturated, making it harder to see clear performance trends. We also exclude AlpacaEval from the comparison because it relies on the GPT API as an evaluator, and conducting a large-scale study would be prohibitively expensive.

## C. Background on Optimal Transport

Optimal Transport (OT) is a principled approach for comparing probability distributions based on their underlying geometry, with strong theoretical guarantees (Villani, 2008). In machine learning, OT has been applied to a variety of domains, including domain adaptation (Courty et al., 2014), generative modeling (Arjovsky et al., 2017), and distance metrics (Alvarez-Melis & Fusi, 2020). The OT problem considers a complete metric space $\mathcal{X}$ with probability measures $\mu, \nu \in \mathcal{P}(\mathcal{X})$, which can be either discrete or continuous. The Kantorovich formulation of the transportation problem is defined as:

$$\text{OT}(\mu, \nu) = \min_{\pi \in \Pi(\mu, \nu)} \int_{\mathcal{X} \times \mathcal{X}} c(x, y) d\pi(x, y)$$

where $c(x, y)$ is a cost function over $\mathbb{R}^+$ and the set of couplings $\Pi(\mu, \nu)$ is defined as the joint probability distributions over the product space $\mathcal{X} \times \mathcal{X}$ with marginals $\mu$ and $\nu$ such that $\Pi(\mu, \nu) = \{\pi \in \mathcal{P}(\mathcal{X} \times \mathcal{X}) | P_{1\#}\pi = \mu, P_{2\#}\pi = \nu\}$. When $\mathcal{X}$ has a given distance metric $d_{\mathcal{X}}$, this is often treated as the cost function such that $c(x, y) = d_{\mathcal{X}}(x, y)^p$ for some $p \geq 1$. This is commonly defined as the $p-$Wasserstein distance where $W_p(\mu, \nu) = \text{OT}(\mu, \nu)^{\frac{1}{p}}$.

In the discrete optimal transport setting with finite samples, as considered in this work, let $\mu$ and $\nu$ be discrete probability measures defined as $\mu = \sum_{i=1}^{n} u_i \delta_{x_i}$ and $\nu = \sum_{j=1}^{m} v_j \delta_{y_j}$, where $\{x_i\}_{i=1}^{n} \in \mathcal{X}^n$ and $\{y_j\}_{j=1}^{m} \in \mathcal{Y}^m$ are points in a metric space (e.g., feature embeddings), with $u \in \mathbb{R}_+^n$, $v \in \mathbb{R}_+^m$, and $\sum_i u_i = \sum_j v_j = 1$. This is now a linear program and can be solved using classical solvers; however, its worst-case complexity is $O(N^3)$, which is often prohibitive in large-scale settings.

### C.1. Entropy-Regularized Optimal Transport

To improve computational efficiency, entropy regularization is commonly applied:

$$\text{OT}(\mu, \nu) = \min_{\pi \in \Pi(\mu, \nu)} \int_{\mathcal{X} \times \mathcal{X}} c(x, y) d\pi(x, y) + \varepsilon \text{H}(\pi | \mu \otimes \nu)$$

where $\text{H}(\pi | \mu \otimes \nu) = \int log(d\pi/d\mu d\nu)$ denotes the entropy of the transport plan. This objective is strongly convex and differentiable, and can be solved efficiently using the Sinkhorn–Knopp algorithm for the discrete case (Sinkhorn, 1964).

## C.2. Unbalanced Optimal Transport

Classical optimal transport assumes that the two measures have equal total mass. In many applications, such as noisy data, out-of-distribution comparison, or dataset selection, this assumption is undesirable. Unbalanced optimal transport relaxes the hard marginal constraints by penalizing deviations from the prescribed marginals (Benamou, 2003; Chizat et al., 2018; Liero et al., 2018; Gabriel & Marco, 2019; S'ejourn'e et al., 2022).

The unbalanced optimal transport problem is defined as

$$\min_{\pi \in \mathcal{M}_+(\mathcal{X} \times \mathcal{X})} \int_{\mathcal{X} \times \mathcal{X}} c(x, y) \, d\pi(x, y) + \lambda_1 \, D(P_{1\#}\pi \,\|\, \mu) + \lambda_2 \, D(P_{2\#}\pi \,\|\, \nu),$$

where $P_{1\#}\pi$ and $P_{2\#}\pi$ denote the first and second marginals of $\pi$, and $D(\cdot\|\cdot)$ is a divergence between non-negative measures, commonly chosen as the Kullback–Leibler divergence.

Adding entropy regularization yields the unbalanced entropy-regularized OT objective:

$$\min_{\pi \in \mathcal{M}_+(\mathcal{X} \times \mathcal{X})} \int_{\mathcal{X} \times \mathcal{X}} c(x, y) \, d\pi(x, y) - \varepsilon H(\pi) + \tau_1 \, \mathrm{KL}(P_{1\#}\pi \,\|\, \mu) + \tau_2 \, \mathrm{KL}(P_{2\#}\pi \,\|\, \nu),$$

where $H(\pi) = -\int \log\left(\frac{d\pi}{dx\,dy}\right) d\pi$ denotes the entropy of the transport plan. As $\tau_1, \tau_2 \to \infty$, the marginal constraints are enforced exactly, recovering the balanced entropy-regularized OT formulation.

# D. Details on LESS

We now describe how LESS computes an influence matrix, which we use as a similarity matrix for our selection algorithms (see Appendix E.3 for implementation details). First, we perform LoRA warmup training of the model $f_{\theta_W}$ on a small randomly sampled warmup set $W \subseteq \mathcal{D}$ for $T$ epochs, saving the model parameters $\{\theta_W^{(t)}\}_{t=1}^T$ along with the corresponding optimizer states at the end of each epoch. Because LLMs are typically trained with Adam, LESS estimates influence via a first-order approximation of Adam training dynamics (Kingma & Ba, 2015), extending earlier first-order influence approximations developed for SGD (Pruthi et al., 2020). Concretely, LESS represents each example using low-dimensional projected features derived from (i) the query gradient and (ii) the candidate Adam update, where the Adam update is a preconditioned gradient computed from Adam's first- and second-moment estimates stored in the optimizer state. To make this scalable, we apply a random projection to these high-dimensional vectors to obtain compact representations in a lower-dimensional subspace. Finally, we compute influence for every query-candidate pair by aggregating cosine similarities between the projected query gradients and the projected candidate Adam updates across checkpoints, weighted by the average learning rate between checkpoints.

More formally, for each checkpoint $t$, we first compute (i) the query gradient and (ii) the candidate Adam update vector:

$$\nabla_\theta \ell\left(\theta_W^{(t)}; q_i\right) \in \mathbb{R}^P,$$

and

$$\Gamma\left(\theta_W^{(t)}; z_j\right) = \frac{\hat{m}^{\,t+1}}{\sqrt{\hat{v}^{\,t+1} + \epsilon}},$$

$$g_t := \nabla_\theta \ell\left(\theta_W^{(t)}; z_j\right),$$

$$\hat{m}^{\,t+1} = \frac{\beta_1 m^t + (1 - \beta_1) g_t}{1 - \beta_1^{t+1}},$$

$$\hat{v}^{\,t+1} = \frac{\beta_2 v^t + (1 - \beta_2) g_t^{\odot 2}}{1 - \beta_2^{t+1}}.$$

Next, we apply a random projection $\Pi \in \mathbb{R}^{P \times d}$ to obtain low-dimensional representations:

$$\tilde{\nabla}\ell\left(\theta_W^{(t)}; q_i\right) = \Pi^\intercal \nabla_\theta \ell\left(\theta_W^{(t)}; q_i\right), \qquad \tilde{\Gamma}\left(\theta_W^{(t)}; z_j\right) = \Pi^\top \Gamma\left(\theta_W^{(t)}; z_j\right).$$

Then the influence of candidate sample $z_j$ on query sample $q_i$ is computed as the weighted cosine similarity aggregated across checkpoints:

$$\text{Inf}_{\text{Adam}}(q_i, z_j) \; = \; \sum_{t=1}^{T} \bar{\eta}_t \; \cos\Big( \tilde{\nabla}\ell\Big(\theta_W^{(t)}; q_i\Big), \tilde{\Gamma}\Big(\theta_W^{(t)}; z_j\Big) \Big). \tag{2}$$

Here, $q_i$ is the $i$-th query sample, $z_j$ is the $j$-th candidate sample, $m^{(t)}$ and $v^{(t)}$ are the first and second moments from the saved checkpoint optimizer states obtained during warmup training, $\beta_1$, $\beta_2$, and $\epsilon$ are Adam-specific hyperparameters, and $\Pi$ is a random projection matrix whose entries are drawn from a Rademacher distribution (i.e., $\Pi_{ij} \sim \mathcal{U}(\{-1, 1\})$) (Johnson & Lindenstrauss, 1984; Park et al., 2023). $\bar{\eta}_t$ is the average learning rate between the $(t-1)$-th and $t$-th checkpoint. We compute $\text{Inf}_{\text{Adam}}(q_i, z_j)$ for all query-candidate pairs to obtain an $\mathbb{R}^{M \times N}$ influence matrix, which we use as the similarity matrix for instruction selection.

## E. Implementation Details: Data Representation

### E.1. RDS+

We use the weighted mean for RDS+ from Ivison et al. (2025) and Muennighoff (2022). For an input sequence $z$ of length $L$, we compute its representation as $z = \sum_{i=1}^{L} w_i h_i$, where $i$ is the token index, $h_i$ is the $i$-th hidden state from the base pre-trained language model, and $w_i = \frac{i}{\sum_{j=1}^{L} j}$ is the positional weight. We truncate both queries and candidates to a maximum length of 2048 tokens.

### E.2. EMBED

Following Ivison et al. (2025), we use GTR-T5 Base (Ni et al., 2022) to get data representations for the candidate pool and query set. For the examples in the query set, we also include an additional prefix `Instruct: Given a sample, find the passages closest to that sample. \nQuery: {query}`.

### E.3. LESS

We closely follow the LESS implementation of Xia et al. (2024). For warmup training, we sample 10,000 instruction–response pairs from the candidate pool and train the base model with LoRA (Hu et al., 2022) using cross-entropy loss. Following the original LESS setup for Llama 2 7B, we apply trainable LoRA parameters to all attention blocks during warmup. We follow the same LoRA setting for Llama 3.2 3B. For the remaining models, following recent recommendations for LoRA training (Schulman & Lab, 2025), we apply trainable LoRA parameters to both the attention and MLP blocks. We use the LoRA hyperparameters from Xia et al. (2024) (rank=128, $\alpha = 512$, dropout=0.1), and all other hyperparameters are provided in Appendix G. We train for 4 epochs and save each checkpoint (total of 316 steps).

Next, we compute gradients of the average loss over response tokens with respect to each LoRA checkpoint to produce the SGD update vectors for the query set and Adam update vectors for the candidate pool. We then apply a random projection using the TRAK package (Park et al., 2023) to map these update vectors into 8192-dimensional feature vectors, resulting in four sets of data representations for both the candidate pool and the query set. For fair comparison with RDS+ and EMBED, we do not average representations when query samples come from the same subtask; instead, we treat them as individual samples. We also compute the average learning rate across training steps between each checkpoint, normalize these values by dividing each by their sum, and finally compute the weighted cosine similarity between candidate and query representations to measure candidate influence on each query sample (Equation 2).

## F. Implementation Details: Selection Algorithms

### F.1. KNN KDE and KNN Uniform

We use the KNN-KDE implementation from TSDS (Liu et al., 2024b) in the released code. We reimplement KNN-Uniform from Algorithm 1 in their paper. We further simplify the implementation by using native PyTorch rather than the FAISS library to compute the distances between data representations. For a fair comparison with other methods, we use cosine distance rather than L2 distance between the query and candidate data representations. In KNN-KDE with LESS, we compute the cosine distance between candidate examples by modifying Equation 2 as $\mathbf{C_{ij}} = 1 - \sum_{t=1}^{T} \bar{\eta}_t \cos(\tilde{\nabla}\Gamma(z_i, f_{\hat{\theta}_t}), \tilde{\nabla}\Gamma(z_j, f_{\hat{\theta}_t}))$.

We closely followed the hyperparameters from Liu et al. (2024b) for KNN-KDE and KNN-Uniform (if the hyperparameter is used) as set $L = 5,000$ (prefetching nearest neighbors), $K_{KDE} = 1,000$ (KDE neighborhood size), $\sigma = 0.75$ (kernel bandwidth), and $C = 5.0$. We set $\alpha = 0.01$ so that at least $10,000$ candidates would have a transported mass greater than zero.

### F.2. Unbalanced OT (UOT)

Algorithm 1 describes the UOT selection algorithm, a new selection algorithm that explicitly solves the unbalanced optimal transport problem to obtain a transport plan, then selects the candidate samples with the highest mass transported. We use the `sinkhorn_unbalanced` implementation from the POT package (Flamary et al., 2021; 2024) to solve the optimization problem. We set $\varepsilon = 0.01$, $\tau_1 = \infty$, and $\tau_2 = 0.0001$. In all our experiments, we compute the cosine distance between the query set and candidate pool and normalize between 0 and 1 by dividing by 2 as follows $\mathbf{C_{ij}} = \left( 1 - \sum_{t=1}^{T} \bar{\eta}_t \cos( \tilde{\nabla}\Gamma(z_i, f_{\hat{\boldsymbol{\theta}}_t}), \tilde{\nabla}\Gamma(z_j, f_{\hat{\boldsymbol{\theta}}_t})) \right) / 2$. We use this normalized cosine distance matrix as our cost matrix.

---

**Algorithm 1** Unbalanced OT Selection

---

**Input:** Candidate pool $\mathcal{D} = \{z_i = (x_i, y_i)\}_{i=1}^{N}$, distance matrix (or cost matrix) between the query set and the candidate pool $\boldsymbol{C} \in \mathbb{R}^{M \times N}$, entropy regularization term $\varepsilon$, marginal relaxation terms $\tau_1$ and $\tau_2$ and budget $B \leq |\mathcal{D}|$.
**Output:** Selected subset $\hat{S}$.
$\Pi \leftarrow \mathrm{UOT}(\boldsymbol{C}, \varepsilon, \tau_1, \tau_2)$                                      // solve UOT (Appendix C) and get the transport plan
$\mathbf{r} \leftarrow \Pi^T \mathbf{1}_M$                                      // candidate masses: $r_j = \sum_{j=1}^{M} \Pi_{i,j}$
$\pi \leftarrow \mathrm{argsort}(-\mathbf{r})$                                      // indices sorted by descending $m_i$
$\hat{S} \leftarrow \emptyset$
**for** $k = 1$ to $B$ **do**
    $\hat{S} \leftarrow \hat{S} \cup \{z_{\pi_k}\}$                                      // select the top-$B$ candidates
**end for**
**return** $\hat{S}$

---

## G. Training Details and Hyperparameters

We follow a standard supervised fine-tuning pipeline and train the base model with a cross entropy loss over the response tokens. During supervised fine-tuning, we set the maximum sequence length to 2048 tokens. For this reason, we remove instruction-response pairs if the response does not appear in 2048 tokens to avoid a zero loss. The preprocessed candidate dataset contains about 198K examples. We apply the chat template to both the candidate and the query set during instruction selection, training, and evaluation, except during zero-shot evaluation of the pre-trained base language model.

| Hyperparameters | Values |
|---|---|
| Learning rate | 2e-5 |
| Learning rate scheduler | linear |
| Number of epochs | 2 |
| Warmup ratio | 0.03 |
| Optimizer | AdamW |
| Adam betas | (0.9, 0.999) |
| Adam epsilon | 1e-8 |
| Weight decay | 0.0 |
| Max. gradient norm | 1.0 |
| Max. sequence length | 2048 |
| Effective batch size | 128 |
| Mixed precision | bf16 |

*Table 3.* Hyperparameters used to train the base models on the selected instructions.

Table 3 lists all the hyperparameters used to train the base models on the selected data. We closely follow the training setup from Ivison et al. (2025). We run all the training and evaluation experiments on a single NVIDIA H100 GPU with 80GB of memory.

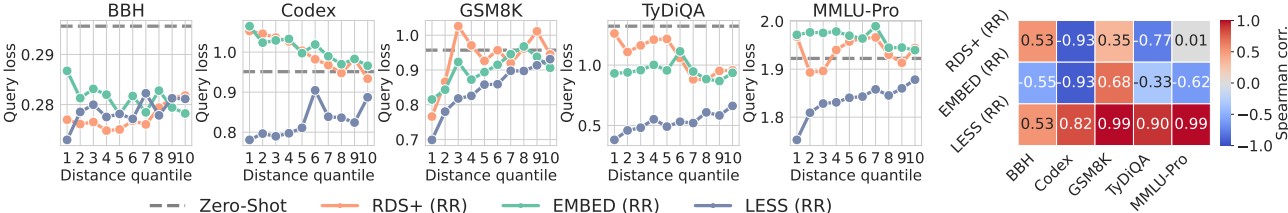

*Figure 9.* **Query loss across subset-query distance sub-quantiles and Spearman correlation.** We further stratify the first distance quantile from Section 5.1 into 10 sub-quantiles (1 = closest, 10 = farthest), select 500 examples per sub-quantile, and train the Llama 2 7B model. We report loss on the query set and Spearman correlation per dataset. LESS (RR) maintains a strong monotonic increase in loss with distance (high Spearman correlation), whereas RDS+ (RR) and EMBED (RR) show weak or inconsistent correlations.

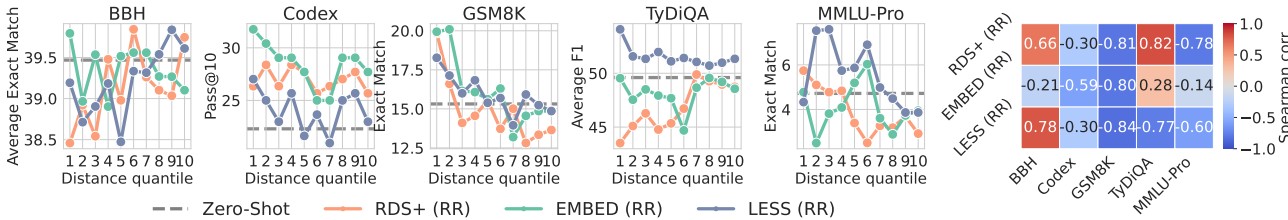

*Figure 10.* **Downstream performance across subset-query distance sub-quantiles and Spearman correlation.** Using the same sub-quantile construction and training protocol as Figure 9, we evaluate downstream task performance across sub-quantiles and report Spearman correlation per dataset (more negative is better). LESS (RR) shows a stronger negative correlation on average, but the performance differences are small, suggesting that many subsets within the closest quantile result in similar downstream performance.

## H. Fine-Grained Stratification of the Nearest Distance Quantile

Building on the distance quantile experiment (Section 5.1), we now aim to determine whether data representations can differentiate between very similar subsets, i.e., subsets that are closer to each other. Since we know the first distance quantile contains the samples most similar to the query set, we further subdivide it into 10 distance quantiles, select the top-K samples from each, and train the base models.

**Setup.** We closely follow the experiment setup from Section 5.1 for creating the distance quantiles, training, and evaluation. We further subdivide the closest distance quantile (first distance quantile) into 10 distance quantiles using the same procedure. Then, we select the top-500 samples from the distance quantiles and use them as training data to train Llama 2 7B.

**Results.** Figure 9 shows that LESS (RR) shows a high Spearman correlation across target tasks with the query loss. On the other hand, the RDS+ (RR) and EMBED (RR) do not show strong correlations across target tasks and often exhibit negative correlations with the distance sub-quantiles, suggesting that these data representations cannot differentiate between similar subsets. Figure 10 further shows that while LESS (RR), on average, shows a stronger negative Spearman correlation on the downstream evaluation compared to the other baselines, the difference in downstream performance appears to be similar. This suggests that multiple subsets in the first distance quantile can achieve a similar downstream performance.

## I. Differences between Selected Subsets

Here, we understand the differences between the selected subsets created using different data representations from Section 5.2. We compute average token length and Jaccard index across the selected subsets for query sets when the budget is 10,000 samples.

Figure 11 shows that subsets created with LESS (RR) contain shorter sequences than those produced by RDS+ (RR) and EMBED (RR) across all target tasks. We suspect this bias toward longer sequences may explain the competitive performance of RDS+ (RR) and EMBED (RR) as we increase the budgets across target tasks (Zhao et al., 2024a). Finally, Figure 12 shows that RDS+ (RR) and EMBED (RR) have higher Jaccard indices compared to LESS (RR), suggesting that model-based

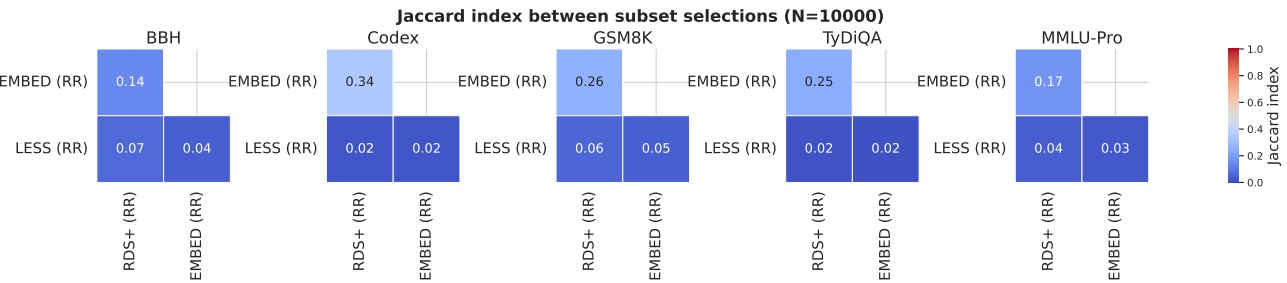

*Figure 11.* **Average token length of the query set and the selected subsets with different data representations.** We find that LESS (RR) is biased towards shorter sequences, whereas RDS+ (RR) and EMBED (RR) select subsets with longer sequences.

*Figure 12.* **Jaccard index between selected subsets created using different data representations.** We find that model-based representations (RDS+ and EMBED) have a higher Jaccard index compared to LESS.

embeddings share greater similarity in the examples they select.

## J. Cheaper Proxies for LESS

LESS data representations are computationally expensive as they require a forward and a backward pass over all the candidate samples. Here, we revisit the experiment of computing the LESS data representations from a smaller proxy model and then use them to select samples from the candidate pool to train the larger model (Appendix D.5 in Xia et al. (2024)). We broaden the set of proxy models, varying in size and include pre-trained LLMs trained with different token budgets to better understand how the performance of the larger base model relates to size and token budgets.

**Setup.** We consider the following proxy models: Pythia 160M (Biderman et al., 2023), SmolLM 135M (Allal et al., 2024), SmolLM2 135M (Allal et al., 2025), and Llama 3.2 3B (Grattafiori et al., 2024). We follow the same procedure to obtain the data representations for all proxy models (Appendix E.3 and Appendix G) and select samples for different budgets using a greedy round-robin approach. Then, we train Llama 2 7B on the selected instructions and report the downstream performance. We also include LESS without the proxy models, along with the Random baseline, to better contextualize the results.

**Results.** Figure 13 shows that, at higher budgets, smaller proxy models outperform the baseline LESS (RR, Llama 2 7B) on three out of the five target tasks. While these results suggest that proxy models for instruction selection are viable, similar to Xia et al. (2024), not all the proxy models perform well. We find that Pythia-160M performs as poorly as Random on several target tasks.

Next, we see that LESS (RR, SmolLM-135M) and LESS (RR, SmolLM2-135M) outperform or match Random across all target tasks, and sometimes even outperform LESS (RR, Llama 2 7B). A key difference between the two is that SmolLM-135M is pre-trained on 600B tokens, whereas SmolLM2-135M is pre-trained on 2 trillion tokens. We observe that on some target tasks, such as BBH and GSM8K, LESS (RR, SmolLM2-135M) achieves higher performance at low budgets, whereas on the rest, LESS (RR, SmolLM-135M) either matches or outperforms LESS (RR, SmolLM2-135M). These results show that the relationship between training tokens and proxy data representation for instruction selection remains unclear.

Finally, we observe that LESS (RR, Llama 3.2 3B) outperforms the baseline LESS (RR, Llama 2 7B) on three out of five target tasks, suggesting a trade-off between proxy and downstream model sizes in how well they approximate instruction selection of the larger models.

Overall, these results suggest that using proxy models for instruction selection to train larger models can dramatically reduce

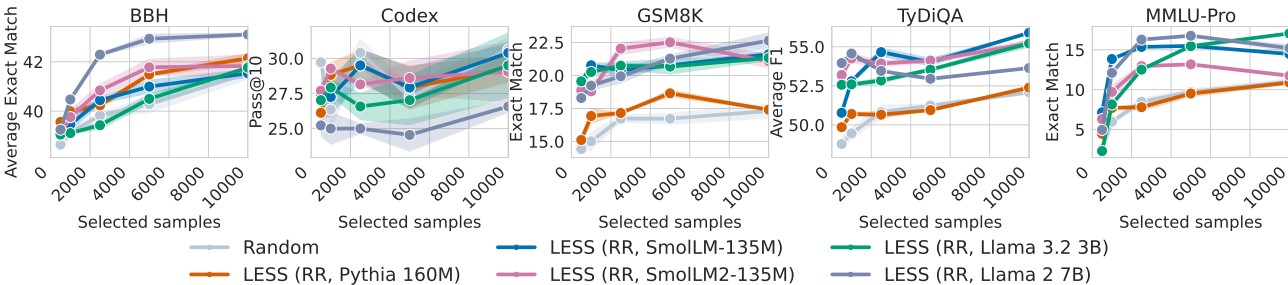

*Figure 13.* **Cheaper proxies for LESS (fixed selection algorithm).** With the same greedy round-robin selection procedure and the budgets from Section 5.2, we report downstream performance of Llama 2 7B when LESS representations are computed using proxy models, averaged across three seeds and the standard error. SmolLM-135M and SmolLM2-135M consistently match or outperform the Random baseline across target tasks, whereas Pythia-160M often matches Random on several tasks. Llama 3.2 3B matches or outperforms LESS computed with Llama 2 7B on multiple target tasks, suggesting a trade-off between proxy and target model size when approximating instruction selection.

cost, but that a more thorough investigation is necessary to understand how to choose proxy models (Khaddaj et al., 2025).

## K. KNN-Uniform and KNN-KDE with L2 Distance

We now select instructions with KNN-Uniform and KNN-KDE selection algorithms and use L2 distances instead of cosine distance to match the original implementation in Liu et al. (2024b).

**Setup.** We use the Llama 2 7B LESS representations for both the query set and the candidate pool to compute distances. Instead of computing the weighted average between the query and the candidate representation (Equation 2), following Liu et al. (2024b), we scale the representation for the epoch checkpoint by the average learning rate for that epoch, concatenate all the scaled representations across epochs, and normalize them by the L2 norm. Then, we compute the L2 distance between the query and the candidate data representations. In KNN-KDE, we use the concatenated candidate data representations to compute L2 distances between them. We follow the same hyperparameters from Appendix K to select the instructions for a given budget. We report the downstream performance of these two selection methods across three seeds. We contextualize these results by comparing them with KNN-KDE and KNN-Uniform when implemented with cosine distance.

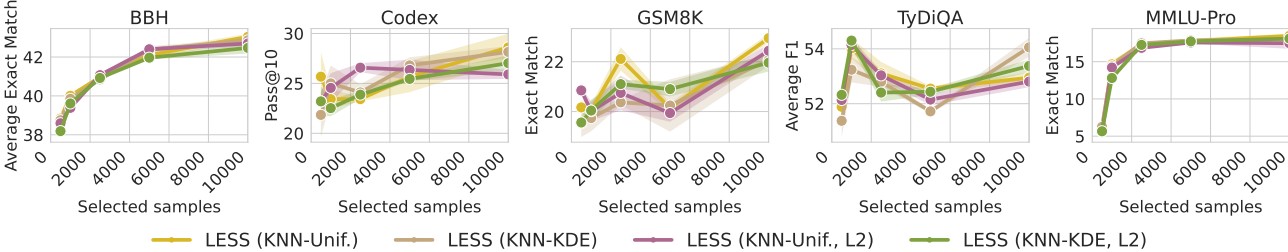

*Figure 14.* **KNN-Uniform and KNN-KDE with L2 distance (fixed data representation).** With the same Llama 2 7B LESS representations and the budgets from Section 5.3, we report downstream performance for KNN-Uniform and KNN-KDE when distances are computed with L2 (following Liu et al. (2024b)), averaged across three seeds and the standard error. We compare against cosine distance variants and find similar performance trends across budgets.

**Results.** Figure 14 shows that both the KNN-KDE and KNN-Uniform with L2 show similar performance trends when cosine distance is used to compute the distances, which further validates our conclusions regarding the choice of selection algorithms in Section 5.3.

# L. Proofs

## L.1. Lemmas used for proofs of Theorem 6.1 and Theorem 6.2

**Lemma L.1** (**Domain adaptation bound**; adapted from Theorem 2 in (Redko et al., 2017))**.** *Let $S$ and $T$ be two samples of size $N_S$ and $N_T$ drawn i.i.d from probability measures $P_S$ and $P_T$ on $\mathbb{R}^d$, respectively. Let $\hat{P}_S := \frac{1}{N_S} \sum_{x_S \in S} \delta_{x_S}$ and $\hat{P}_T := \frac{1}{N_T} \sum_{x_T \in T} \delta_{x_T}$, where $\delta_x$ is the Dirac measure at $x$, be the associated empirical measures. Let $\ell$ be any loss function that is symmetric, convex, bounded, obeys the triangular inequality, and for $z = (x, y)$ has the parametric form $|y - f_\theta(x)|^q$ for some $q > 0$. Let $\Theta$ denote the model's parameter space, and for any dataset $D$ let $\theta_D \in \Theta$ denote the parameters obtained by training on $D$. Then, for any $d' > d$ and $c' < \sqrt{2}$, there exists some constant $N_0$ depending on $d'$ such that for any $\delta > 0$ and $\min(N_S, N_T) \geq N_0 \max(\delta^{-(d'+2)}, 1)$, with probability at least $1 - \delta$, the following holds for every dataset $D$:*

$$L_T(\theta_D) \leq L_S(\theta_D) + W_1(\hat{P}_S, \hat{P}_T) + \sqrt{\frac{2}{c'} \log\left(\frac{1}{\delta}\right)} \left(\frac{1}{\sqrt{N_S}} + \frac{1}{\sqrt{N_T}}\right) + \lambda_{S,T} \tag{3}$$

where $\lambda_{S,T} := \min_{\theta \in \Theta}\{L_T(\theta) + L_S(\theta)\}$ is the minimum combined error over all model parameters.

### Assumptions for Lemma L.2 and Theorem 6.2

A1. (**Strong convexity**) For any fixed dataset $D \subset \mathcal{Z}$, the empirical risk $L_D(\theta)$ is $\mu$–strongly convex in $\theta$.

A2. (**Smooth data dependence**) The gradient $\nabla_\theta \ell(z; \theta)$ is $G_{\theta z}$-Lipschitz with respect to the data point $z$, i.e., $\|\nabla_\theta \ell(\theta; z) - \nabla_\theta \ell(\theta; z')\| \leq G_{\theta z}\|z - z'\|$, for all $\theta$ in a relevant neighborhood.

A3. (**Lipschitzness at the ERM**) For a fixed dataset $D$, the empirical risk $L_D$ is $K$-Lipschitz with respect to the model parameter. That is, for any parameters $\theta$ and $\theta'$, $|L_D(\theta) - L_D(\theta')| \leq K\|\theta - \theta'\|$.

**Lemma L.2** (**Wasserstein stability bound**)**.** *Let $\mathcal{T}$ be a target dataset. Assume assumptions A1–A3 hold. Then, for any two datasets $D$ and $D'$, $|L_\mathcal{T}(\theta_D) - L_\mathcal{T}(\theta_{D'})| \leq C_\star W_1(\hat{P}_D, \hat{P}_{D'})$ with $C_\star := \frac{KG_{\theta z}}{\mu}$.*

*Proof.* Let $\theta_D = \arg\min_\theta L_D(\theta)$ and $\theta_{D'} = \arg\min_\theta L_{D'}(\theta)$. Using strong convexity once at $L_D$ (Assumption A1)

$$\mu\|\theta_D - \theta_{D'}\| \leq \|\nabla L_D(\theta_D) - \nabla L_D(\theta_{D'})\| = \|\nabla L_D(\theta_{D'})\|$$

where the last equality comes from $\|\nabla L_D(\theta_D)\| = 0$ due to optimality. Similarly, because $\|\nabla L_{D'}(\theta_{D'})\| = 0$, then

$$\mu\|\theta_D - \theta_{D'}\| \leq \|\nabla L_D(\theta_{D'}) - \nabla L_{D'}(\theta_{D'})\|. \tag{4}$$

For any fixed $\theta$, Assumption A2 and Kantorovich–Rubinstein duality (Arjovsky et al., 2017) gives

$$\|\nabla_\theta L_D(\theta) - \nabla_\theta L_{D'}(\theta)\| \leq G_{\theta z} W_1(\hat{P}_D, \hat{P}_{D'}). \tag{5}$$

Combining (4) and (5)

$$\|\theta_D - \theta_{D'}\| \leq \frac{G_{\theta z}}{\mu} W_1(\hat{P}_D, \hat{P}_{D'}). \tag{6}$$

Assumption A3 gives $|L_\mathcal{T}(\theta_D) - L_\mathcal{T}(\theta_{D'})| \leq K\|\theta_D - \theta_{D'}\|$, which combined with (6) gives

$$|L_\mathcal{T}(\theta_D) - L_\mathcal{T}(\theta_{D'})| \leq K\|\theta_D - \theta_{D'}\| \leq \frac{K \cdot G_{\theta z}}{\mu} W_1(\hat{P}_D, \hat{P}_{D'}). \tag{7}$$

$\square$

**Lemma L.3 (High-probability bound for the Wasserstein distance of an empirical measure).** *Let $\mathcal{D} = \{z_1, \ldots, z_N\} \subset \mathbb{R}^d$ lie in a set of diameter $\Delta$ and $d \geq 3$. Let $\mathcal{S}^{\mathrm{rnd}} := (s_1, \ldots, s_B)$ be a tuple of $B$ elements sampled i.i.d. uniformly from $\mathcal{D}$, and let $\hat{P}_{\mathcal{D}}$ and $\hat{P}_{\mathcal{S}^{\mathrm{rnd}}}$ be the empirical distributions on $\mathcal{D}$ and $\mathcal{S}^{\mathrm{rnd}}$ respectively. Then there exists a constant $C_d > 0$ depending only on $d$ (with $q = 2$ fixed) such that, for any $\delta \in (0, 1)$, with probability at least $1 - \delta$ over the draw of $\mathcal{S}^{\mathrm{rnd}}$:*

$$W_1(\hat{P}_{\mathcal{S}^{\mathrm{rnd}}}, \hat{P}_{\mathcal{D}}) \leq C_d \Delta B^{-1/d} + \Delta \sqrt{\frac{\log(1/\delta)}{2B}} \tag{8}$$

*Proof.* The proof proceeds by decomposing $W_1(\hat{P}_{\mathcal{S}^{\mathrm{rnd}}}, \hat{P}_{D_c})$ into its expectation and its concentration around the mean. First, we use Theorem 1 in Fournier & Guillin (2015) with $p = 1$ and $q = 2$, applied on bounded support: since $\mathrm{diam}(\mathrm{supp}(\hat{P}_{\mathcal{D}})) \leq \Delta$, we have $M_2(\hat{P}_{\mathcal{D}})^{1/2} \leq \Delta$. This theorem yields $\mathbb{E}[W_1(\hat{P}_{S_{\mathrm{rand}}}, \hat{P}_c)] \leq C(1, d, 2)\Delta (B^{-1/d} + B^{-1/2})$, and because $d \geq 3$ implies $B^{-1/2} \leq B^{-1/d}$, we can absorb the $B^{-1/2}$ term into the constant, defining $C_d := 2C(1, d, 2)$. Then, for dimensions $d \geq 3$:

$$\mathbb{E}[W_1(\hat{P}_{\mathcal{S}^{\mathrm{rnd}}}, \hat{P}_{\mathcal{D}})] \leq C_d \Delta B^{-1/d} \tag{9}$$

where $C_d$ is a constant depending only on dimension $d$.

Next, we view $f(\mathcal{S}^{\mathrm{rnd}}) := W_1(\hat{P}_{\mathcal{S}^{\mathrm{rnd}}}, \hat{P}_{\mathcal{D}})$ as a function of the random subset $\mathcal{S}^{\mathrm{rnd}}$. Since the data lies in a bounded domain of diameter $\Delta$, changing a single data point in $\mathcal{S}^{\mathrm{rnd}}$ changes the probability mass at that location by $1/B$, and moves it by a distance of at most $\Delta$. Therefore, the function $f$ satisfies the bounded difference property with constant $c_i = \Delta/B$. Then, by McDiarmid's inequality (McDiarmid, 1989), for any $\epsilon > 0$:

$$\mathbb{P}\left(W_1(\hat{P}_{\mathcal{S}^{\mathrm{rnd}}}, \hat{P}_{\mathcal{D}}) - \mathbb{E}[W_1(\hat{P}_{\mathcal{S}^{\mathrm{rnd}}}, \hat{P}_{\mathcal{D}})] \geq \epsilon\right) \leq \exp\left(\frac{-2\epsilon^2}{\sum_{i=1}^B (\Delta/B)^2}\right) = \exp\left(\frac{-2B\epsilon^2}{\Delta^2}\right) \tag{10}$$

Similar arguments have been used, for example, by Solomon et al. (2022), who apply McDiarmid's inequality to the Wasserstein distance viewed as a function of two random datasets, taking expectations over both. In contrast, in our setting, we fix $\mathcal{D}$ and treat the Wasserstein distance as a function of the random subset $\mathcal{S}^{\mathrm{rnd}}$, taking the expectation over a single dataset. In this case, the bounded-differences condition required by McDiarmid's inequality follows directly from the stronger assumption that the support of $\mathcal{D}$, and hence the support of $\mathcal{S}^{\mathrm{rnd}} \subseteq \mathcal{D}$, is bounded with diameter at most $\Delta$.

Setting the right hand side in (10) to $\delta$ and solving for $\epsilon$, we get $\epsilon = \Delta \sqrt{\frac{\log(1/\delta)}{2B}}$. Thus, with probability at least $1 - \delta$:

$$W_1(\hat{P}_{\mathcal{S}^{\mathrm{rnd}}}, \hat{P}_{\mathcal{D}}) \leq \mathbb{E}[W_1(\hat{P}_{\mathcal{S}^{\mathrm{rnd}}}, \hat{P}_{\mathcal{D}})] + \Delta \sqrt{\frac{\log(1/\delta)}{2B}} \tag{11}$$

Combining the expectation bound (9) with the concentration bound (11) yields the final result:

$$W_1(\hat{P}_{\mathcal{S}^{\mathrm{rnd}}}, \hat{P}_{\mathcal{D}}) \leq C_d \Delta B^{-1/d} + \Delta \sqrt{\frac{\log(1/\delta)}{2B}} \tag{12}$$

$\square$

### L.2. Theorem 6.1

Let $\ell : \Theta \times \mathcal{Z} \to \mathbb{R}_+$ be any loss function that is symmetric, convex, bounded, satisfies the triangle inequality, and for $z = (x, y)$ admits the parametric form $\ell(\theta; z) = |y - f_\theta(x)|^q$ for some $q > 0$. Let $\mathcal{D}$ denote a labeled candidate pool, and let $\mathcal{S} \subseteq \mathcal{D}$ be any subset of size $B := |\mathcal{S}|$. Let $\mathcal{Q}$ (the query set) and $\mathcal{T}$ (the test set) be labeled datasets. Then for any $c' < \sqrt{2}$, with probability at least $1 - 2\delta$:

$$L_{\mathcal{T}}(\theta_{\mathcal{S}}) \leq \underbrace{W_1(\hat{P}_{\mathcal{S}}, \hat{P}_{\mathcal{Q}})}_{\substack{\text{Subset and query} \\ \text{dataset distance}}} + \underbrace{W_1(\hat{P}_{\mathcal{Q}}, \hat{P}_{\mathcal{T}})}_{\substack{\text{Query and test} \\ \text{dataset distance}}} + \underbrace{L_{\mathcal{S}}(\theta_{\mathcal{S}})}_{\text{training error}} + \zeta \sqrt{\frac{2}{c'} \log\left(\frac{1}{\delta}\right)} + \tilde{\lambda}$$

where $W_1$ is the 1-Wasserstein distance with respect to the underlying metric on the embedding space $\mathcal{Z}$, $\zeta$ is a constant determined by the size of the datasets, with $\zeta = B^{-\frac{1}{2}} + 2|\mathcal{Q}|^{-\frac{1}{2}} + |\mathcal{T}|^{-\frac{1}{2}}$, and $\tilde{\lambda}$ is the combined error of the dataset $\tilde{\mathcal{S}} \subseteq \mathcal{D}$ of size $B$ that minimizes the combined error of $L_{\mathcal{S}}(\theta_{\tilde{\mathcal{S}}}) + 2L_{\mathcal{Q}}(\theta_{\tilde{\mathcal{S}}}) + L_{\mathcal{T}}(\theta_{\tilde{\mathcal{S}}})$.

*Proof.* The proof proceeds by applying Lemma L.1 twice. We first apply it to $L_{\mathcal{T}}(\theta_{\mathcal{S}})$, treating $\mathcal{T}$ as the target dataset and $\mathcal{Q}$ as the source dataset, with the function obtained by training on $\mathcal{S}$. We then apply it to $L_{\mathcal{Q}}(\theta_{\mathcal{S}})$, now treating $\mathcal{Q}$ as the target dataset and $\mathcal{S}$ as the source dataset, again using the prediction function trained on $\mathcal{S}$. This yields:

$$L_{\mathcal{T}}(\theta_{\mathcal{S}}) \le L_{\mathcal{Q}}(\theta_{\mathcal{S}}) + W_1(\hat{P}_{\mathcal{Q}}, \hat{P}_{\mathcal{T}}) + \sqrt{\frac{2}{c'} \log\left(\frac{1}{\delta}\right)} \left(\frac{1}{\sqrt{|\mathcal{Q}|}} + \frac{1}{\sqrt{|\mathcal{T}|}}\right) + \lambda_{\mathcal{T},\mathcal{Q}} \tag{13}$$

$$\le L_{\mathcal{S}}(\theta_{\mathcal{S}}) + W_1(\hat{P}_{\mathcal{S}}, \hat{P}_{\mathcal{Q}}) + W_1(\hat{P}_{\mathcal{Q}}, \hat{P}_{\mathcal{T}}) + \sqrt{\frac{2}{c'} \log\left(\frac{1}{\delta}\right)} \left(\frac{1}{\sqrt{B}} + \frac{2}{\sqrt{|\mathcal{Q}|}} + \frac{1}{\sqrt{|\mathcal{T}|}}\right) + \lambda_{\mathcal{T},\mathcal{Q}} + \lambda_{\mathcal{Q},\mathcal{S}} \tag{14}$$

$$\le L_{\mathcal{S}}(\theta_{\mathcal{S}}) + W_1(\hat{P}_{\mathcal{S}}, \hat{P}_{\mathcal{Q}}) + W_1(\hat{P}_{\mathcal{Q}}, \hat{P}_{\mathcal{T}}) + \zeta\sqrt{\frac{2}{c'} \log\left(\frac{1}{\delta}\right)} + \lambda_{\mathcal{T},\mathcal{Q}} + \lambda_{\mathcal{Q},\mathcal{S}} \tag{15}$$

where $\lambda_{\mathcal{T},\mathcal{Q}} := \min_{\theta \in \Theta} L_Q(\theta) + L_T(\theta)$ and $\lambda_{\mathcal{Q},\mathcal{S}} := \min_{\theta \in \Theta} L_S(\theta) + L_Q(\theta)$ and $\zeta = B^{-\frac{1}{2}} + 2|\mathcal{Q}|^{-\frac{1}{2}} + |\mathcal{T}|^{-\frac{1}{2}}$.

Since the sum of pointwise minima is bounded above by the minimum of the sum: $\lambda_{\mathcal{T},\mathcal{Q}} + \lambda_{\mathcal{Q},\mathcal{S}} \le \min_{\theta \in \Theta}\{L_{\mathcal{S}}(\theta) + 2L_{\mathcal{Q}}(\theta) + L_{\mathcal{T}}(\theta)\} =: \lambda$. We refer to $\lambda$ as the minimum combined error over the full parameter space $\Theta$.

The minimum above is taken over all $\theta \in \Theta$. In practice, we are restricted to parameters obtainable by training on some subset $\mathcal{S} \subseteq \mathcal{D}$, i.e., parameters of the form $\theta_{\mathcal{S}}$. Restricting the minimum to this set can only increase it: $\lambda = \min_{\theta \in \Theta}\{L_{\mathcal{S}}(\theta) + 2L_{\mathcal{Q}}(\theta) + L_{\mathcal{T}}(\theta)\} \le \min_{\mathcal{S} \subseteq \mathcal{D}}\{L_{\mathcal{S}}(\theta_{\mathcal{S}}) + 2L_{\mathcal{Q}}(\theta_{\mathcal{S}}) + L_{\mathcal{T}}(\theta_{\mathcal{S}})\} =: \tilde{\lambda}$. We denote by $\tilde{\mathcal{S}}$ the subset achieving this minimum, so that $\tilde{\lambda}$ is the combined error attained by training on $\tilde{\mathcal{S}}$. Substituting back, we obtain:

$$L_{\mathcal{T}}(\theta_{\mathcal{S}}) \le L_{\mathcal{S}}(\theta_{\mathcal{S}}) + W_1(\hat{P}_{\mathcal{S}}, \hat{P}_{\mathcal{Q}}) + W_1(\hat{P}_{\mathcal{Q}}, \hat{P}_{\mathcal{T}}) + \zeta\sqrt{\frac{2}{c'} \log\left(\frac{1}{\delta}\right)} + \lambda \tag{16}$$

$$\le L_{\mathcal{S}}(\theta_{\mathcal{S}}) + W_1(\hat{P}_{\mathcal{S}}, \hat{P}_{\mathcal{Q}}) + W_1(\hat{P}_{\mathcal{Q}}, \hat{P}_{\mathcal{T}}) + \zeta\sqrt{\frac{2}{c'} \log\left(\frac{1}{\delta}\right)} + \tilde{\lambda} \tag{17}$$

$\square$

## L.3. Theorem 6.2

Let the candidate pool $\mathcal{D} \subset \mathbb{R}^d$ lie in a set of diameter $\Delta$ and $d \ge 3$. Let $\mathcal{S}^{\mathrm{rnd}} := (s_1, \ldots, s_B)$ be a tuple of $B$ elements sampled i.i.d. uniformly from $\mathcal{D}$, and let $S_W^* \subseteq \mathcal{D}$ be a subset of size $B$ that minimizes the 1-Wasserstein distance to the query set $\mathcal{Q}$, i.e., $\mathcal{S}_W^* \subset \arg\min_{\mathcal{S} \subseteq \mathcal{D}; |\mathcal{S}| = B} W_1(\hat{P}_{\mathcal{S}}, \hat{P}_{\mathcal{Q}})$. Assume assumptions A1–A3 hold. Then, there exists a constant $C_d > 0$, depending only on the dimension $d$, such that, with probability at least $1 - 2\delta$:

$$L_{\mathcal{T}}(\theta_{\mathcal{S}^{\mathrm{rnd}}}) - L_{\mathcal{T}}(\theta_{S_W^*}) \le C_\star \left( \underbrace{C_d \Delta B^{-1/d}}_{\substack{\text{Curse of} \\ \text{dimensionality}}} + \underbrace{\Delta\sqrt{\frac{\log(1/\delta)}{2B}}}_{\text{Concentration bound}} + \underbrace{W_1(\hat{P}_{\mathcal{D}}, \hat{P}_{\mathcal{Q}})}_{\text{Pool-query mismatch}} + \underbrace{W_1(\hat{P}_{S_W^*}, \hat{P}_{\mathcal{Q}})}_{\text{Distance residual}} \right)$$

*Proof.*

$$L_{\mathcal{T}}(\theta_{\mathcal{S}^{\mathrm{rnd}}}) - L_{\mathcal{T}}(\theta_{S_W^*}) \le C_\star W_1(\hat{P}_{\mathcal{S}^{\mathrm{rnd}}}, \hat{P}_{S_W^*}) \tag{18}$$

$$\le C_\star \left( W_1(\hat{P}_{\mathcal{S}^{\mathrm{rnd}}}, \hat{P}_{\mathcal{Q}}) + W_1(P_{S_W^*}, \hat{P}_{\mathcal{Q}}) \right) \tag{19}$$

$$\le C_\star \left( W_1(\hat{P}_{\mathcal{S}^{\mathrm{rnd}}}, \hat{P}_{\mathcal{D}}) + W_1(\hat{P}_{\mathcal{D}}, \hat{P}_{\mathcal{Q}}) + W_1(P_{S_W^*}, \hat{P}_{\mathcal{Q}}) \right) \tag{20}$$

$$\le C_\star \left( C_d \Delta B^{-1/d} + \Delta\sqrt{\frac{\log(1/\delta)}{2B}} + W_1(\hat{P}_{\mathcal{D}}, \hat{P}_{\mathcal{Q}}) + W_1(\hat{P}_{\mathcal{S}_W^*}, \hat{P}_{\mathcal{Q}}) \right) \tag{21}$$

$\square$

Line 18 follows from applying Lemma L.2, line 19 applied triangle inequality by introducing the distance to the empirical distribution of $\mathcal{Q}$, line 20 again uses triangle inequality by introducing the distance to the empirical distribution of $\mathcal{D}$, and 21 is a result of applying Lemma L.3 on $W_1(\hat{P}_{\mathcal{S}^{\mathrm{rnd}}}, \hat{P}_{\mathcal{D}})$.

## M. Model Ablations with Tulu V2

We run ablations for experiments in Section 5 across four models of different sizes: Llama 3.2 3B (Grattafiori et al., 2024), SmolLM3 3B (Bakouch et al., 2025), Qwen3 4B Base (Yang et al., 2025), and Olmo 3 7B (Olmo et al., 2025).

Below, we summarize our key takeaways from the experiment across models (See Figures 15, 16, 17, and 18):

**Distance quantile experiment.**    Only LESS (RR) creates subsets whose quantile distances strongly correlate with the loss. However, with over-trained models such as SmolLM3 3B Base and Qwen3 4B Base, the performance trends for LESS (RR) on datasets such as BBH and Codex show less correlation with the distance quantiles.

**Effect of data representation across subset budgets.**    Although no single data representation with a fixed selection algorithm selects subsets that always perform the best, LESS with greedy round robin performs the best across all models on most of the target tasks compared to other model-based representation baselines. But on some datasets and models (MMLU Pro with Qwen3 4B Base), we observe that the zero-shot baseline outperforms even instruction-tuned models, suggesting that either the candidate pool may not have sufficient training examples to further improve the performance, or due to an accidental leakage of the target task in the pre-training corpus (Olmo et al., 2025).

**Effect of selection algorithm across subset budgets.**    We observe that the performance trends for different selection algorithm generalize to other models. For instance, LESS (RR) achieves the strongest results on BBH with Llama 3.2 3B and Qwen3 4B Base. We also find that, at higher budgets, often LESS (UOT) delivers the best MMLU-Pro performance with Olmo 3 7B Base. Overall, these findings indicate that, for certain target tasks, selection algorithms tend to exhibit consistent performance across different models.

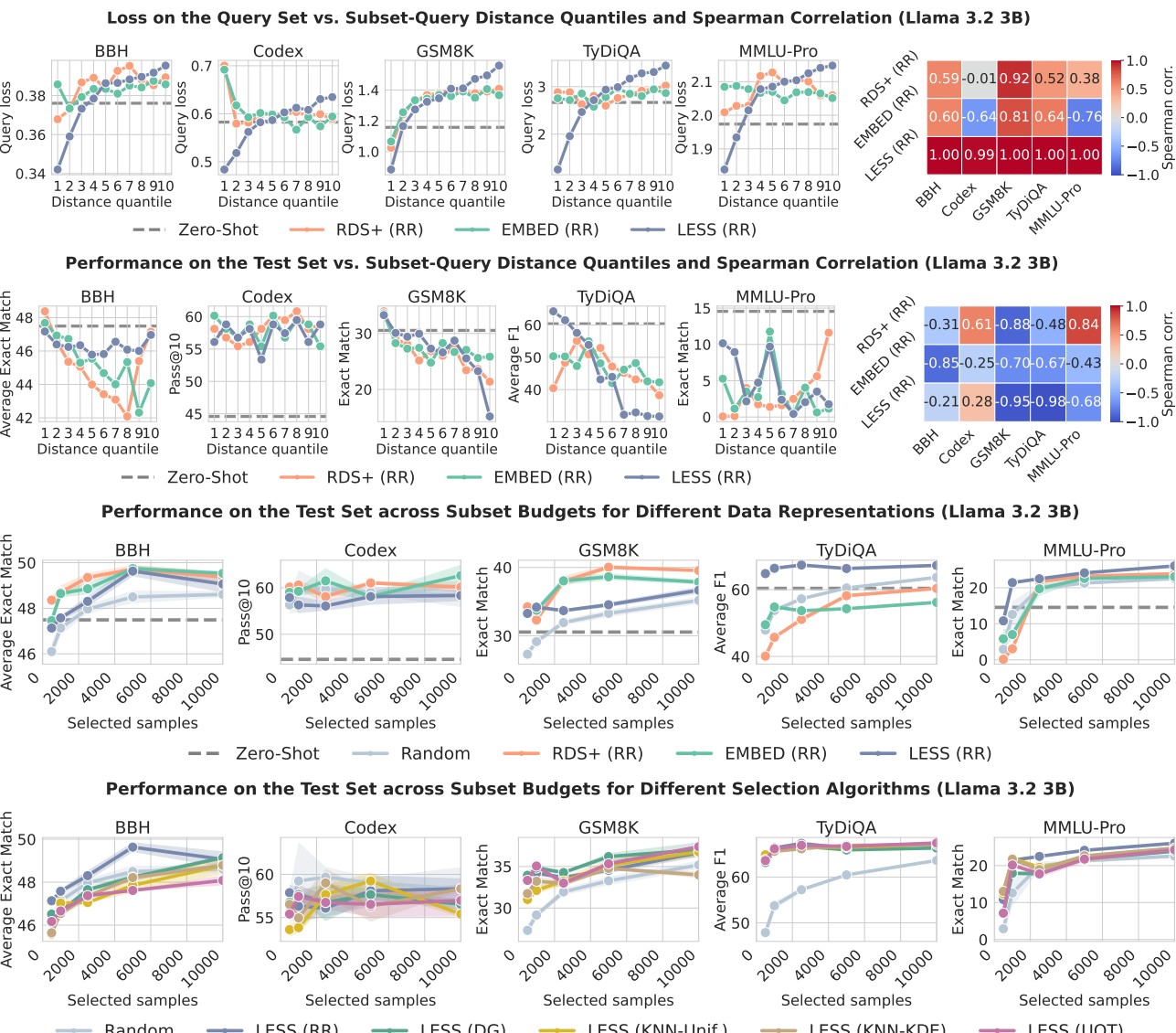

*Figure 15.* Ablation experiments with Llama 3.2 3B.

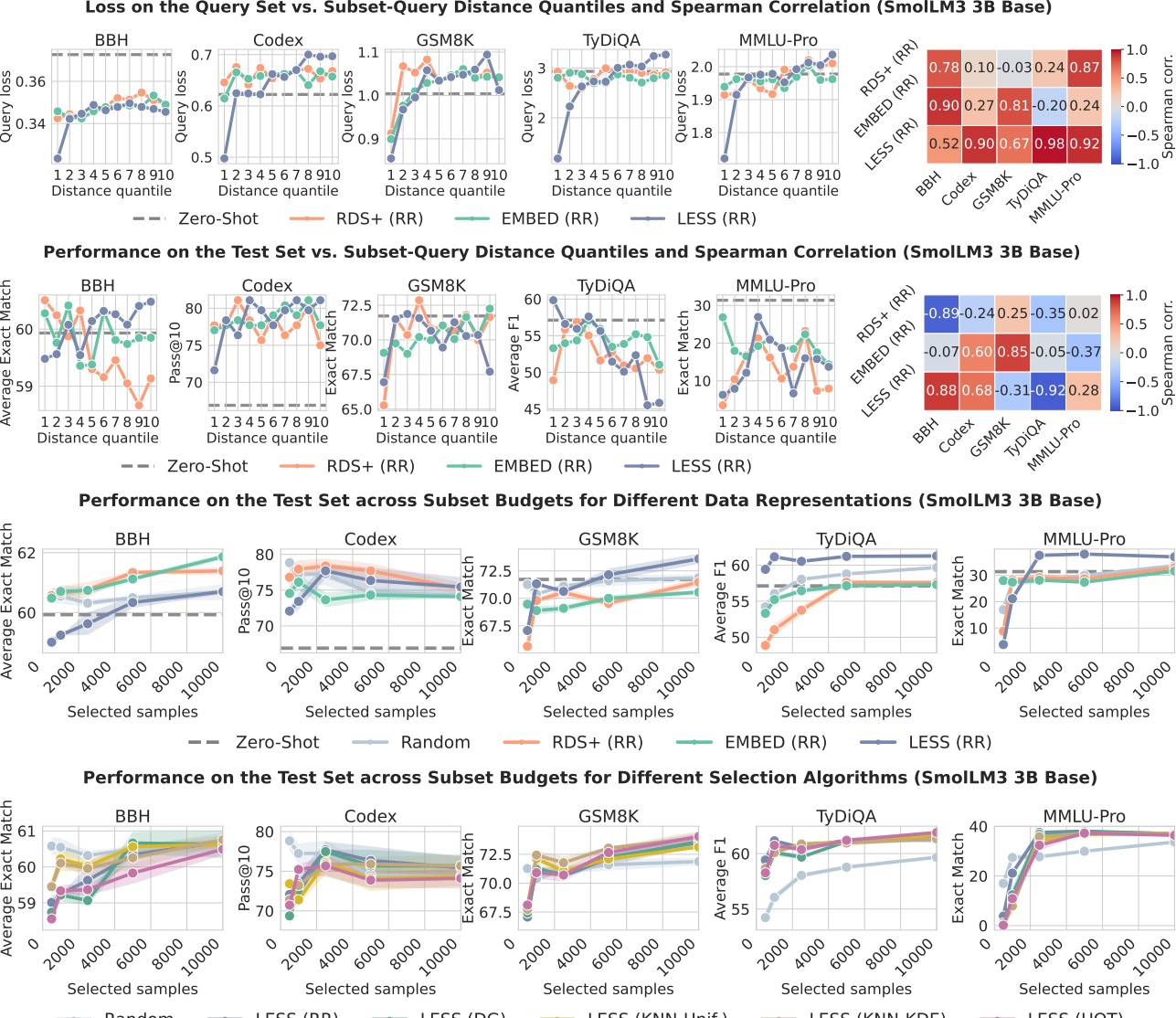

*Figure 16.* Ablation experiments with SmolLM3 3B.

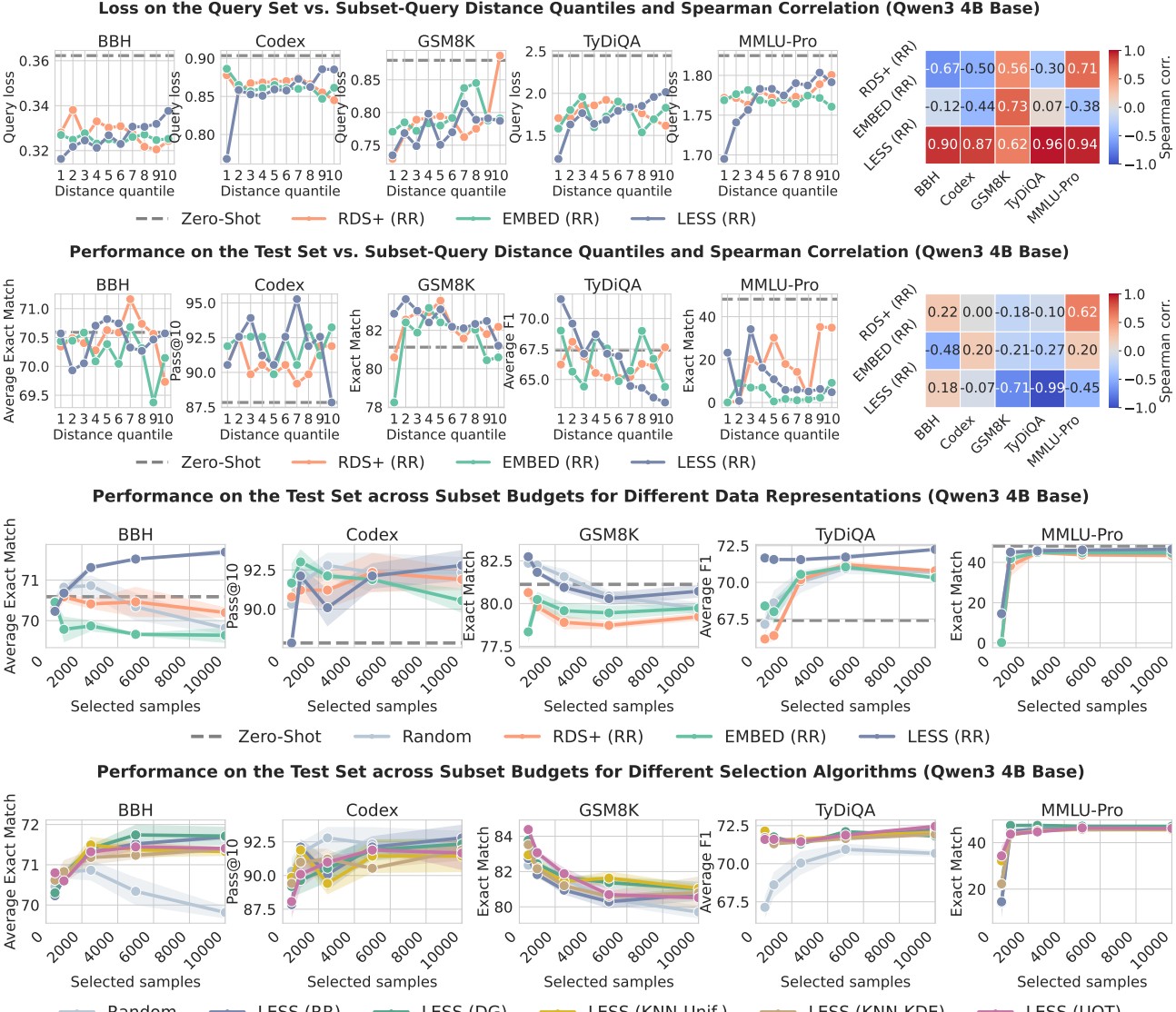

*Figure 17.* Ablation experiments with Qwen3 4B.

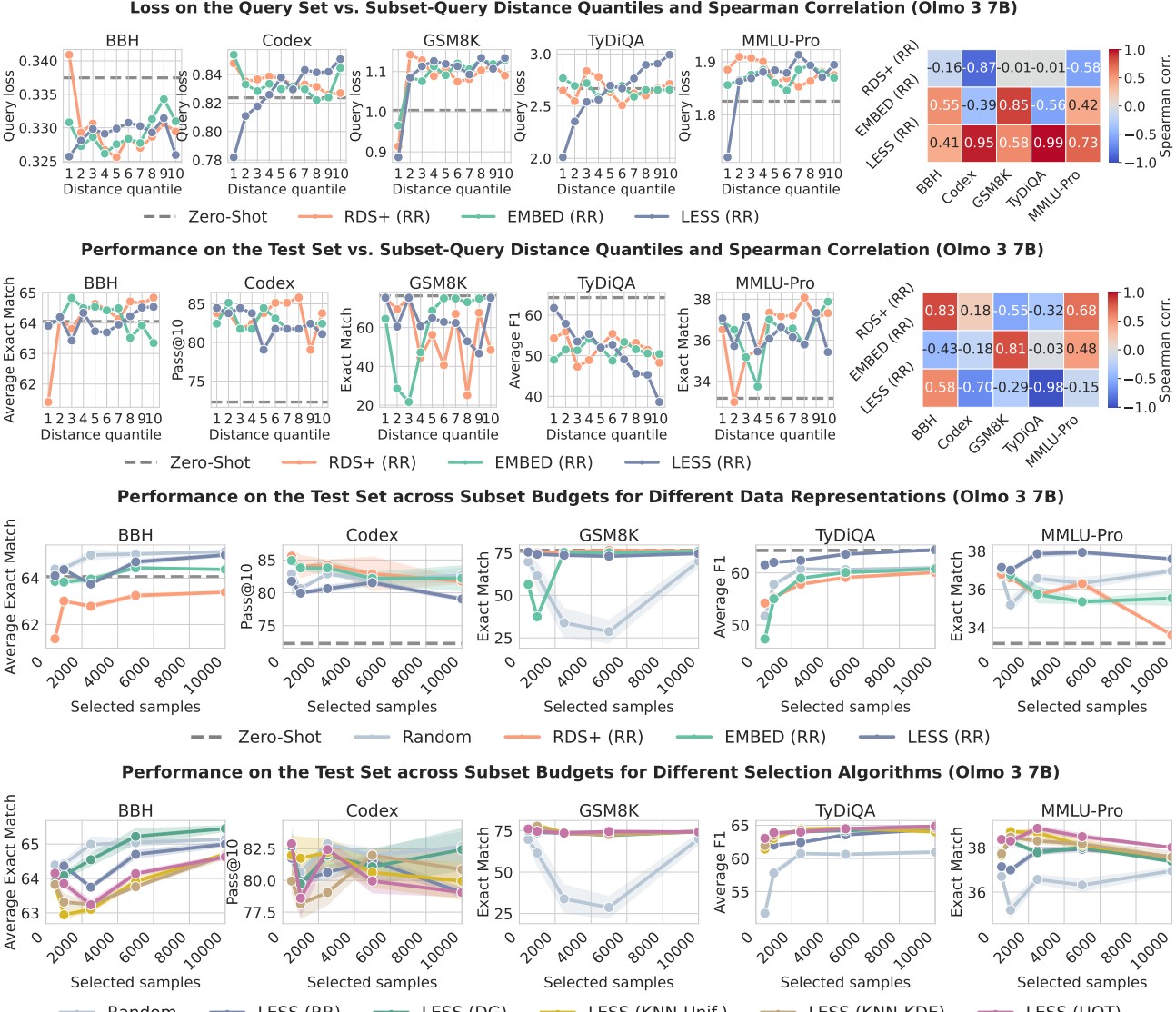

*Figure 18.* Ablation experiments with Olmo 3 7B.

# N. Multitask Instruction Selection

Here, we experiment with multitask instruction selection, where the goal is to select instructions from the candidate pool for multiple target tasks rather than a single task.

**Setup.** We jointly select instructions for all target tasks described in Appendix B. For each data representation, we concatenate the single-task similarity matrices across all target tasks, treat the resulting matrix as a single multitask selection problem, select instructions using different selection algorithms, and train models using the same procedure described in Section 4.

**Results.** Figure 19 shows the multitask instruction selection results for Llama 3.2 3B. We find that the results closely follow those from the single-task selection experiments with Llama 3.2 3B (Figure 15), suggesting that insights from single-task targeted instruction selection generalize to multitask instruction selection.

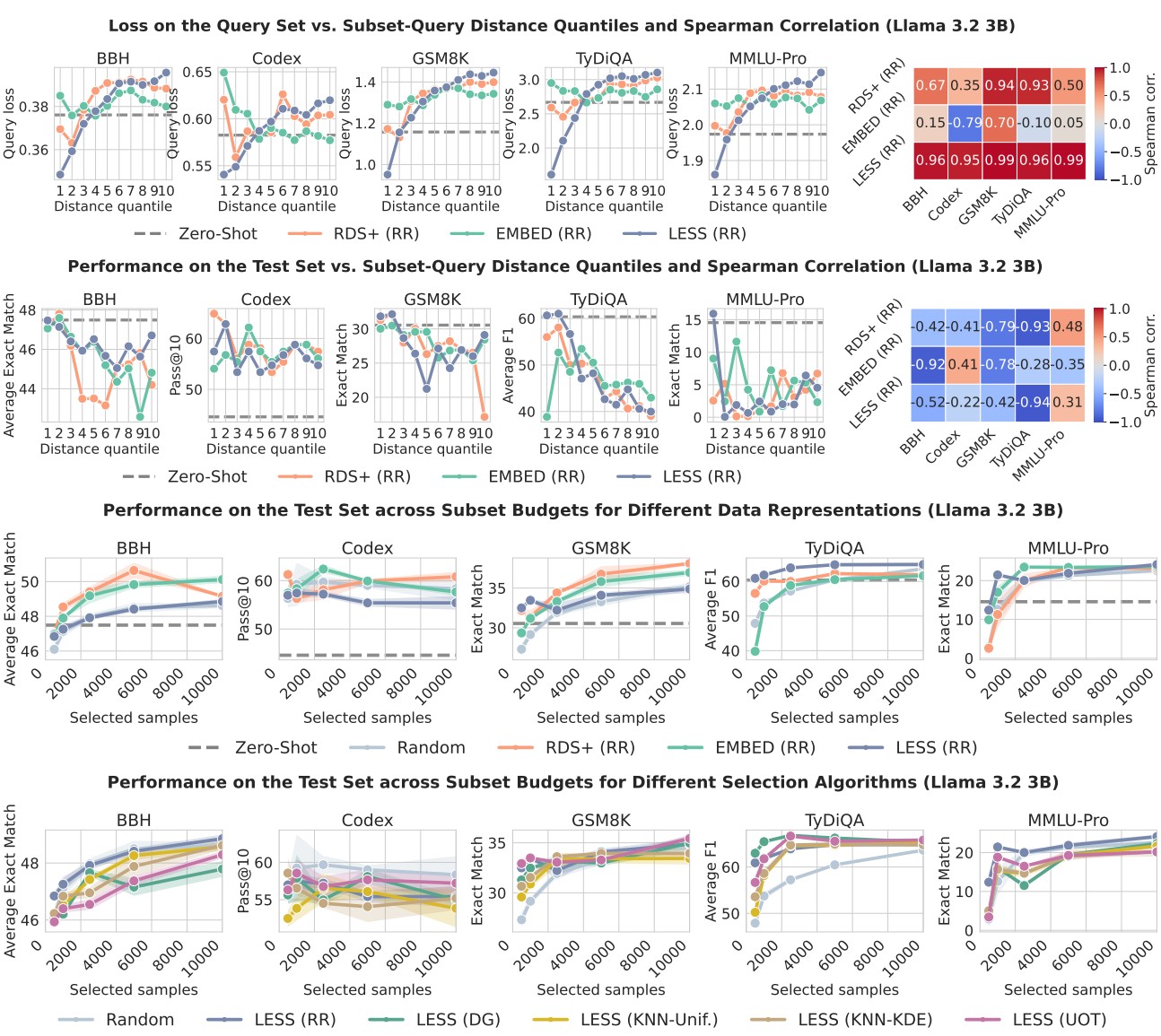

*Figure 19.* Multitask instruction selection experiments with Llama 3.2 3B

# O. Model Ablations with Dolci Instruct

We experiment with Dolci Instruct (Olmo et al., 2025) as the candidate pool to better understand targeted instruction selection with a different candidate pool. Dolci Instruct is a recent post-training dataset that builds on the earlier Tulu V2 (Ivison et al., 2023) and Tulu V3 (Lambert et al., 2025) mixtures while broadening their coverage to include synthetic and curated instructions for mathematical reasoning, code generation, precise instruction following, tool use, multilingual tasks, and safety-related behaviors.

**Setup.** We filter the Dolci-Instruct-SFT dataset by removing all the instructions that require tool use and randomly sample $200,000$ examples from the dataset. We use this dataset as our candidate pool. For the rest of the setup, we follow the same experimental setup from Section 4. Since the candidate pool contains longer sequences, we increase the max. sequence length to 4096 tokens during training, including intermediate data representation computation. We repeat all the experiments from Section 5 with Llama 2 7B (Touvron et al., 2023), Llama 3.2 3B (Grattafiori et al., 2024), SmolLM3 3B (Bakouch et al., 2025), Qwen3 4B Base (Yang et al., 2025), and Olmo 3 7B (Olmo et al., 2025).

**Results.** We now briefly summarize the results across Figures 20, 21, 22, 23, and 24. First, consistent with our findings, we find that LESS creates subsets whose distance to the query strongly correlates with the query loss across model-task pairs. However, we also observe that the subset distance to the query often strongly correlates with the downstream metric, with LESS showing a stronger correlation than other data representations. Next, in contrast to our experiments with Tulu V2, we observe that RDS+ (RR) and EMBED (RR) often match or outperform LESS (RR) on BBH, Codex, and GSM8K. On the other hand, LESS (RR) consistently outperforms the other methods on TyDiQA across models. Then, we observe that no selection algorithm with LESS dominates across datasets, budgets, or models. Finally, we observe that Random is a competitive baseline at higher budgets, highlighting the benefits of a well-curated candidate pool such as Tulu V2 and Dolci Instruct. These results underscore a broader candidate-pool dependence in targeted instruction selection: while LESS remains a reliable signal for query loss under Dolci-Instruct, downstream gains and method rankings are substantially less stable across tasks, models, and budgets.

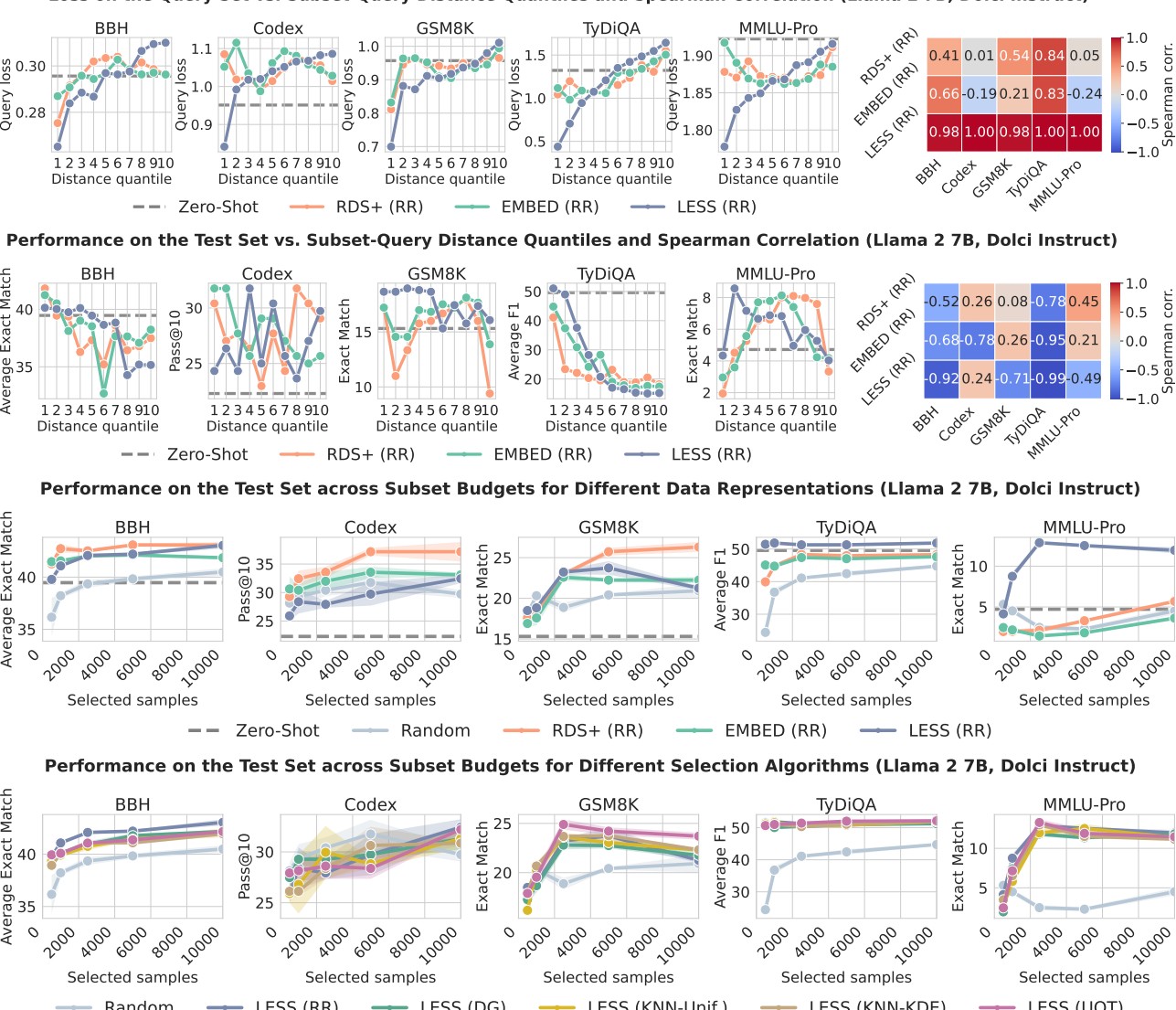

*Figure 20.* Ablation experiments with Llama 3.2 3B using Dolci Instruct as the candidate pool.

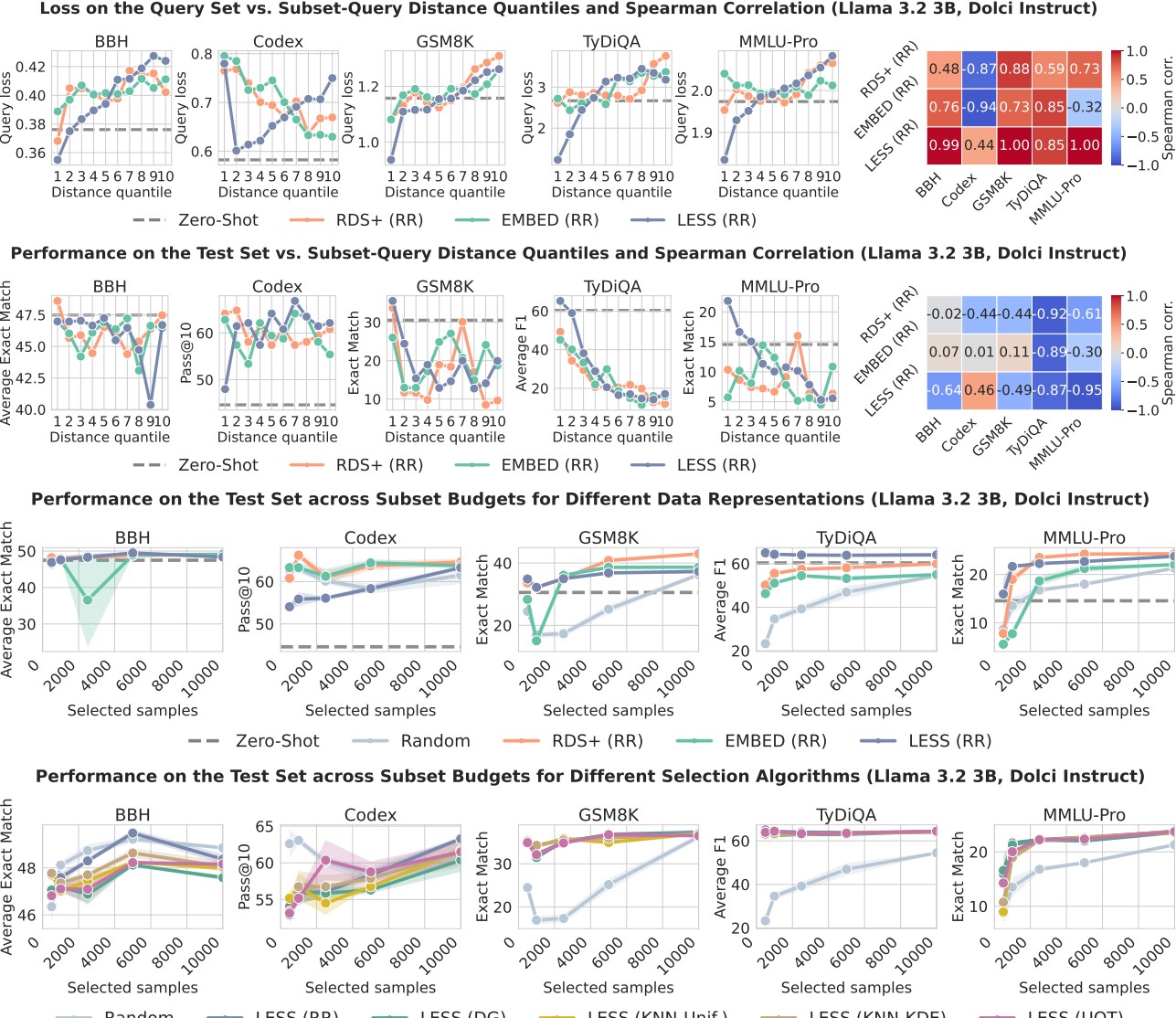

*Figure 21.* Ablation experiments with Llama 3.2 3B using Dolci Instruct as the candidate pool.

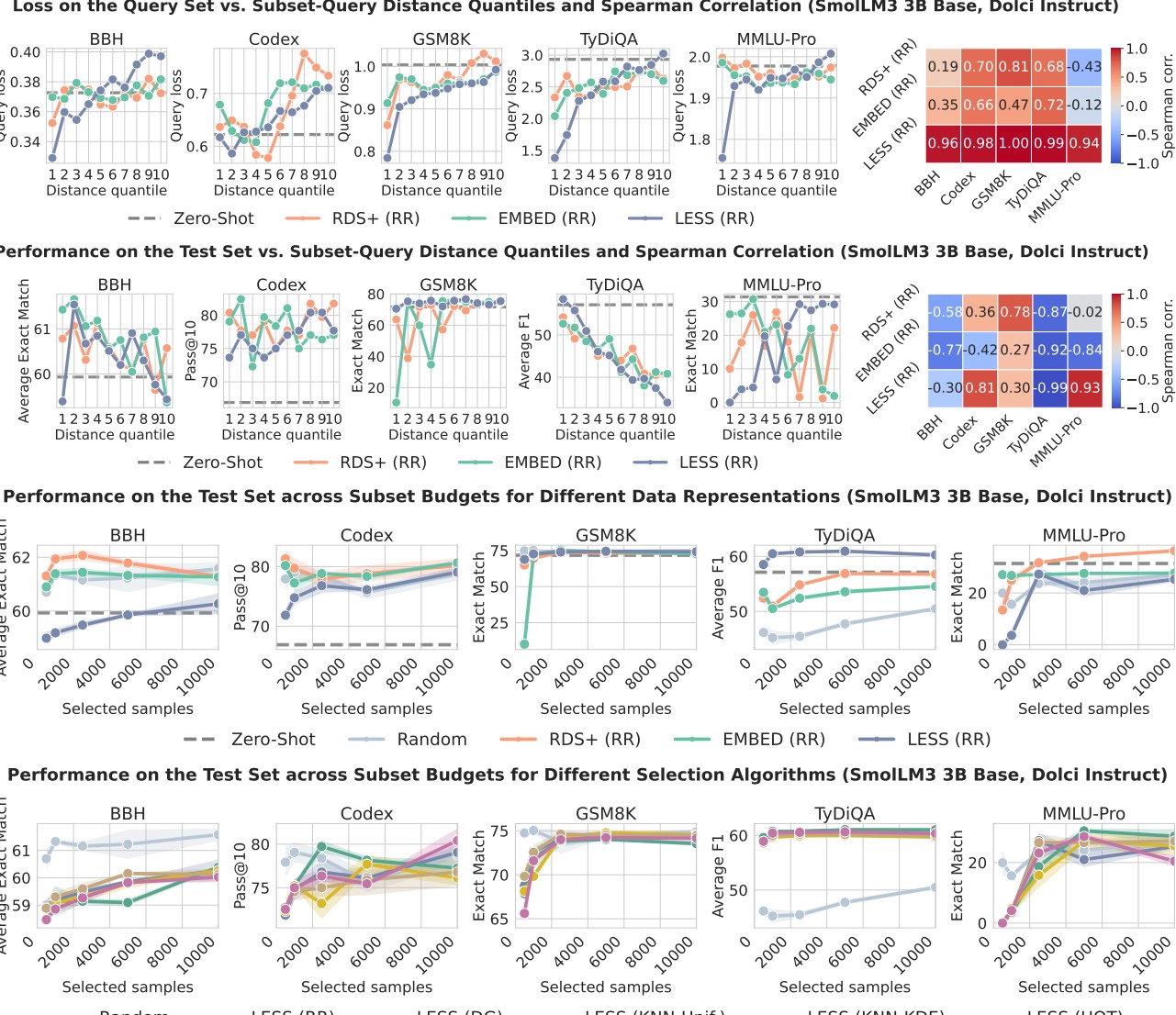

*Figure 22.* Ablation experiments with SmolLM3 3B using Dolci Instruct as the candidate pool.

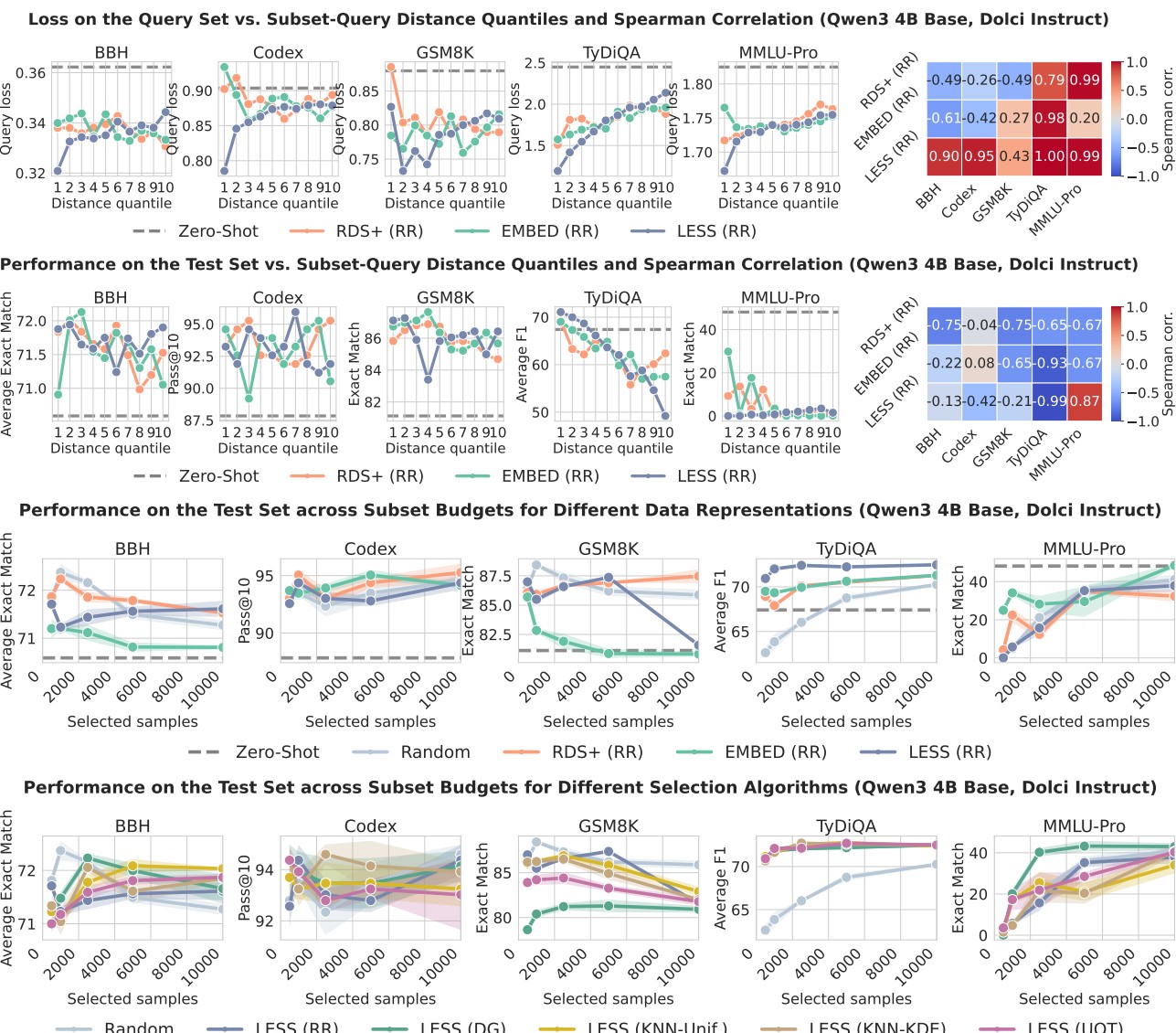

*Figure 23.* Ablation experiments with Qwen3 4B using Dolci Instruct as the candidate pool.

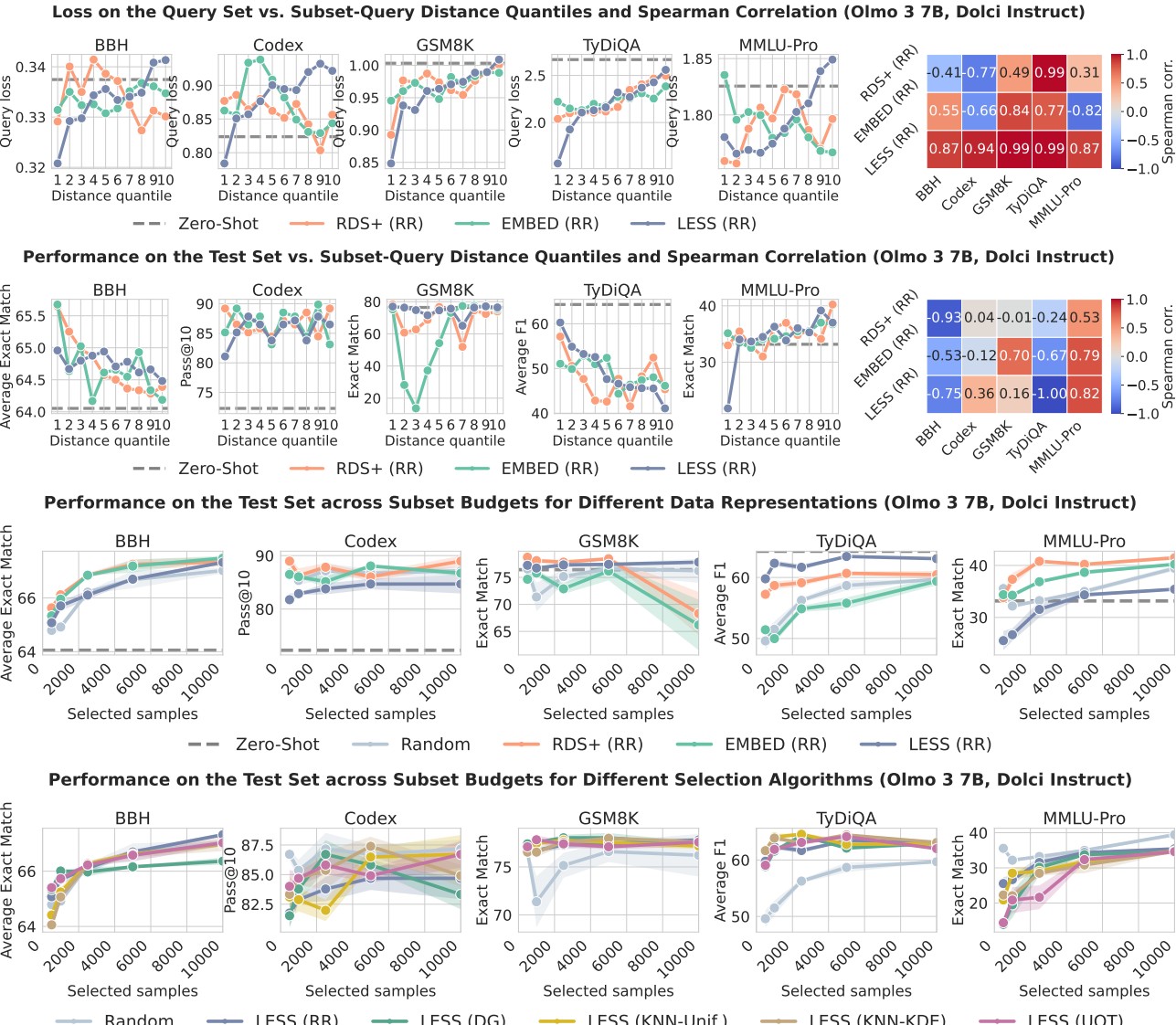

*Figure 24.* Ablation experiments with Olmo 3 7B using Dolci Instruct as the candidate pool.

