# OpenReview forum: "A Critical Look at Targeted Instruction Selection: Disentangling What Matters (and What Doesn’t)"
_ICML.cc/2026/Conference — ICML 2026 regular_

### Official Review · Reviewer_a3Pf · 2026-02-22

**Soundness:** 3
**Presentation:** 3
**Significance:** 3
**Originality:** 2
**Overall Recommendation:** 4
**Confidence:** 3

**Summary:**

This paper proposes a decoupling framework that separates target instruction selection into two independent variables, data representation (LESS, RDS+, EMBED) and selection algorithm (RR, DG, KNN-Uniform, KNN-KDE, UOT), and systematically evaluates all combinations across five target tasks. The core findings are: only LESS representations yield subset-to-query distances that reliably predict performance; greedy round-robin is optimal at low budgets; and random sampling frequently matches or outperforms selection methods at larger budgets.

**Compliance With Llm Reviewing Policy:**

Affirmed.

**Final Justification:**

The intrinsic dimension estimate of roughly 45 clarifies the vacuous bound issue, the clarification that prior work did not actually compare LESS at the 5.8M scale corrects my misunderstanding, and the Dolci-Instruct-SFT experiment adds needed diversity. The core disconnect between query loss and downstream performance, where EMBED+RR outperforms LESS+RR on 4 of 5 tasks, remains open, but the authors acknowledge this and frame it as motivation for future work. I prefer to maintain my score.

**Key Questions For Authors:**

Q1. EMBED+RR outperforms LESS+RR on downstream metrics in 4 of 5 tasks despite much higher query loss. Does this indicate that query loss is an unreliable proxy for downstream performance? What is the practical implication for the distance-minimization framework in Section 6?

Q2. Prior work at larger scale (5.8M candidate pool, 2.5M selected) finds RDS+ superior to LESS-based methods, contradicting this paper's conclusion. Are the findings here specific to small-scale candidate pools? Is there a crossover point as pool size grows?

Q3. In Theorem 6.2, the bound scales as $B^{-1/d}$ with $d = 8192$, making the bound essentially vacuous. What is the effective intrinsic dimensionality of the LESS feature space? Is the strong convexity assumption reasonable for LoRA fine-tuning?

Q4. The query sets are very small—16 samples for Codex and 8 for GSM8K—making Wasserstein distance estimates highly unstable. Would larger query sets narrow the gap between query loss and downstream performance and yield more reliable distance estimates?

**Strengths And Weaknesses:**

Strengths

1. The decoupling framework addresses a genuine gap in the literature. Existing methods vary simultaneously along representation, metric, algorithm, and candidate pool dimensions, making comparisons difficult. Isolating each component by holding others fixed is methodologically sound and yields interpretable results.
2. The distance-quantile experiment is well-designed: stratifying candidates by distance and training on each stratum separately cleanly reveals the monotone correlation between LESS distance and query loss, and the disconnect for RDS+/EMBED, providing a mechanistic explanation for why random sampling is often competitive.

Weaknesses

1. The disconnect between query loss and downstream performance is left unexplained. Figures 2–3 show that LESS+RR achieves the lowest query loss, yet EMBED+RR yields higher downstream performance on 4 of 5 tasks. This directly challenges the premise of the Section 6 unifying framework—that minimizing subset-to-query distance improves target performance—and the paper acknowledges but does not analyze the discrepancy.

---

> ### Author Rebuttal · Authors · 2026-03-30
>
> Thank you for the thoughtful feedback on our manuscript. Below we respond to your comments:
>
>
> > Q1. EMBED+RR outperforms LESS+RR on downstream metrics in 4 of 5 tasks despite much higher query loss. Does this indicate that query loss is an unreliable proxy … What is the practical implication for the distance-minimization framework…
>
> The distance-minimization framework described in Section 6 provides a principled theoretical basis for understanding the relationship between subset-query distance and target performance in terms of loss. This means that the notions of distances should correlate more strongly with loss and downstream performance. However, in practice, we find that while LESS correlates strongly with query loss, it might not correspond to the best downstream model. On the other hand, we find that EMBED and RDS+ show very little correlation across datasets, which makes the optimization challenging. These discrepancies **motivate the need for better data representations** whose notions of distance correlate more strongly with query loss and downstream performance. We will make this motivation clear in the final version of the paper.
>
> > Q2. Prior work at larger scale (5.8M candidate pool, 2.5M selected) finds RDS+ superior to LESS-based methods, contradicting this paper's conclusion. …findings here specific to small-scale candidate pools? Is there a crossover point ..
>
> We believe there is a misunderstanding about [a] as it **does not compare LESS and RDS+ at the 5.8M scale**. Figure 1 in [a] explicitly states, “We do not run LESS with 5.8M samples due to its high compute cost,” and therefore does not establish that RDS+ is superior to LESS in that regime. In addition, our findings show that the original LESS configuration, i.e., LESS with doubly greedy (DG), is often not the strongest way to use LESS; in Section 5.3, replacing DG with greedy round-robin materially improves LESS, especially at smaller budgets. You are also right that we observe a crossover or convergence between Random and RDS+ (and other methods). For instance, our current results in Figure 5 already suggest such behavior on MMLU-Pro and TyDiQA, where the gap between RDS+ and Random narrows and, in some cases, reverses as the budget increases.
>
> > Q3. In Theorem 6.2, the bound scales as $B^{-1/d}$ with $d=8192$, making the bound essentially vacuous. …effective intrinsic dimensionality … feature space? Is the strong convexity assumption …
>
> Our intent with Theorem 6.2 is for the bound to be qualitative rather than quantitative, and to provide intuition for why the gains of targeted selection over a random baseline diminish as the budget increases. We agree that the current bound has an important limitation: the projected LESS feature space dimension $d$ can be very high ($d=8192$ in our case), which makes the resulting $B^{-1/d}$ rate quite loose. However, the **intrinsic dimension of these features can be much lower**, which could further tighten the bound. To investigate this, we empirically estimate the intrinsic dimension of the LESS features from Llama 3.2 3B using the MLE estimator [b] from the skdim library [c]. We uniformly sample 10,000 features, estimate the intrinsic dimension using `n_neighbors=20` across five seeds, and report the mean and standard deviation. We find that the intrinsic dimension is 45.24 +/- 4.62, which is significantly smaller than the dimension of the feature space. Moreover, related theoretical work [d] shows that Wasserstein convergence rates can depend on intrinsic notions of dimension, such as Wasserstein dimension, suggesting a possible route to tighter future bounds in our setting. Finally, we agree that the strong convexity assumption may not be realistic for LoRA fine-tuning in practice, which we will explicitly clarify in the next final version of the paper.
>
> > Q4. The query sets are very small… distance estimates highly unstable. Would larger query sets …yield more reliable distance estimates?
>
> We agree that **very small query sets can yield noisier loss estimates** and a weaker match between query loss and downstream performance. This aligns with Figures 4–5: BBH and MMLU-Pro use larger query sets and show clearer trends, whereas Codex and GSM8K are noisier. **Query set size is not the only factor**, however. Although TyDiQA also uses a small query set, its samples cover the evaluation languages, likely making it more representative of the target distribution and leading to a stronger correlation.
>
> **References**
>
> [a] Large-Scale Data Selection for Instruction Tuning, arxiv 25.
>
> [b] Translated poisson mixture model for stratification learning, IJCV 08.
>
> [c] Scikit-dimension: a python package for intrinsic dimension estimation, Entropy 21.
>
> [d] Sharp asymptotic and finite-sample rates of convergence of empirical measures in Wasserstein distance, Bernoulli 19.

---

> > ### Author Rebuttal · Reviewer_a3Pf · 2026-04-02
> >
> > Thank you for the rebuttal. The authors have addressed several of my concerns. I maintain my score.

---

> > > ### Author Response · Authors · 2026-04-02
> > >
> > > Thank you for your review and consideration. We would be happy to answer any other concerns that you may have regarding our work.

---

### Official Review · Reviewer_KxA1 · 2026-03-07

**Soundness:** 3
**Presentation:** 2
**Significance:** 3
**Originality:** 3
**Overall Recommendation:** 4
**Confidence:** 3

**Summary:**

This paper presents a systematic study of key components in current approaches to target instruction selection. The authors empirically examine the effects of three data representation methods, five selection algorithms, and five data budgets. Their results show that gradient-based representations such as LESS generally perform well across different settings, while the optimal selection algorithm varies depending on the data budget.

The paper also provides a theoretical perspective on target instruction selection by formulating it as a subset–query distance minimization problem. It further shows that the benefits of target instruction selection diminish as the data budget increases, which is consistent with the empirical findings.

**Compliance With Llm Reviewing Policy:**

Affirmed.

**Final Justification:**

My concerns are addressed, and I would like to keep my positive rating.

**Key Questions For Authors:**

See questions in the Weaknesses section.

**Limitations:**

This paper does not have a limitation section.

**Strengths And Weaknesses:**

**Strengths**:

1. The empirical study is very thorough, covering three data representation methods, five selection algorithms, five data budgets, five LLMs for instruction tuning, four proxy models for LESS, and five target benchmarks.

2. The study systematically isolates different components of target instruction selection, enabling a clearer understanding of the pros and cons of different design choices and which combinations perform well under various settings.

3. The results show that zero-shot inference and random selection can be quite competitive in certain settings, raising useful questions about when target data selection is actually necessary.

**Weaknesses**:
1. In Sec. 2, the target instruction selection problem is formulated as minimizing the expected loss on the target task. However, this definition is questionable because training loss is often poorly correlated with downstream task performance. In fact, the empirical results in the paper already suggest such a mismatch, which weakens the theoretical framing.

2. The candidate dataset used for selection, TULU v2 (released in 2023), is outdated, and the study considers only this single dataset. This raises concerns about whether the conclusions would generalize to more recent SFT datasets, given the distribution shifts. For example, many newer Chain-of-Thought (CoT) SFT datasets include substantially longer reasoning traces, whereas responses in TULU v2 are generally much shorter.

3. Each benchmark is considered independently in the study. However, in practice, people often aim to improve LLM performance across multiple tasks. This raises concerns about whether the conclusions still hold when the target is a mixture of tasks, such as distinct tasks (e.g., general knowledge + math reasoning) or somewhat correlated tasks (e.g., code reasoning + math reasoning).

**Minor Weaknesses**:

4. The experiment results in Sec. 5 and Appendix H–M contain many details, and they are not easy to read. Readability could be improved by explicitly highlighting the key takeaway for each subsection.

---

> ### Author Rebuttal · Authors · 2026-03-30
>
> Thank you for the thoughtful feedback on our manuscript. Below we respond to your comments:
>
> > 1. …problem is formulated as minimizing the expected loss on the target task. However, this definition is questionable … a mismatch, which weakens the theoretical framing.
>
> We would like to clarify that we expect the **query loss**, rather than the training loss on the selected subset, **to be more informative about downstream performance**. You are right that our problem is formulated as minimizing the expected loss on the target task. Since we do not have access to the target task during selection, we use the query set as a proxy, assuming it is drawn from the same distribution as the target task. We agree, however, that this proxy is imperfect in practice, and our empirical results already reflect this limitation. This does not weaken our theoretical framing, because Theorem 6.1 explicitly captures this source of mismatch through the query-target discrepancy term $W_1(\hat{P}_Q, \hat{P}_T)$. Furthermore, our formulation is an idealized objective on the true target task, while the query set is only a finite observable proxy. As a result, the theory is naturally conditional on how well the query distribution approximates the true target distribution.
>
> > …the study considers only this single dataset. …
>
> Thank you for your suggestion. We experiment with Tulu v2 to stay as close as possible to [a] in terms of experimental setup without introducing additional confounders. As suggested by you and the **Reviewers YcfY and 8Z96**, we run a preliminary experiment with Dolci-Instruct-SFT [b], a newly proposed dataset from December 2025, used to train Olmo3 Instruct models, which contains longer reasoning traces, as the candidate pool. Since our target tasks did not include tool use tasks, we filtered them and randomly sampled 200K from the dataset as the candidate pool. We follow the same experimental setup and implementation details for the baselines. We train Olmo 3 7B on the selected datasets for one seed.
>
> Our results show that the query loss remains strongly correlated with subset-query distance quantiles computed using LESS. However, we find that RDS+ (RR) outperforms LESS (RR) on most of the datasets, despite being only weakly correlated with the loss. We also observe that Random is a strong baseline and even outperforms LESS (RR) on MMLU Pro. Finally, as in our experiments with Tulu v2, we find that UOT outperforms RR at larger budgets.  We will conduct a comprehensive evaluation and incorporate these new insights into the final version.
>
> **Distance Quantile Experiment.**
> | Method | BBH | Codex | GSM8K | TyDiQA | MMLU-Pro |
> |---|---|---|---|---|---|
> | RDS+ (RR) | -0.45 | -0.75 | 0.22 | 0.98 | 0.13 |
> | EMBED (RR) | 0.49 | -0.64 | 0.84 | 0.77 | -0.90 |
> | LESS (RR) | **0.87** | **0.94** | **0.99** | **0.99** | **0.87** |
>
> **Effect of Data Representation across Subset Budgets.**
> | Method | BBH | Codex | GSM8K | TyDiQA | MMLU-Pro |
> |---|---|---|---|---|---|
> | Budgets | 500/5K/10K | 500/5K/10K | 500/5K/10K | 500/5K/10K | 500/5K/10K |
> | Random | 64.6/66.5/66.9 | 85.8/**88.5**/87.2 | 76.5/75.7/73.8 | 49.1/59.7/59.8 | **35.6**/35.3/38.6 |
> | RDS+ (RR) | **65.6**/**66.9**/67.0 | **88.5**/87.2/87.2 | **78.2**/78.2/74.8 | 57.0/61.2/61.6 | 33.1/**40.4**/**40.7** |
> | EMBED (RR) | 65.4/**66.9**/**67.6** | 87.8/**88.5**/86.5 | 75.1/**78.7**/58.5 | 51.4/55.6/60.2 | 34.8/38.3/40.3 |
> | LESS (RR) | 65.0/66.8/67.4 | 79.7/86.5/**87.8** | 77.2/77.0/**77.6** | **59.8**/**63.5**/**62.9** | 22.0/35.0/35.7 |
>
>
> **Effect of Selection Algorithm across Subset Budgets**
> | Method | BBH | Codex | GSM8K | TyDiQA | MMLU-Pro |
> |---|---|---|---|---|---|
> | Budgets | 500/5K/10K | 500/5K/10K | 500/5K/10K | 500/5K/10K | 500/5K/10K |
> | LESS (RR) | 65.0/**66.8**/**67.4** | 79.7/86.5/**87.8** | 77.2/77.0/77.6 | 59.8/63.5/62.9 | **22.0**/**35.0**/**35.7** |
> | LESS (DG) | **65.3**/66.2/66.2 | 83.1/85.1/81.8 | **77.6**/**78.9**/76.9 | 59.4/62.0/62.3 | 15.6/33.9/34.9 |
> | LESS (KNN-Unif.) | 64.2/66.7/66.8 | **86.5**/**87.8**/84.5 | 77.2/76.8/77.0 | **61.6**/62.2/**63.2** | 18.4/30.3/35.0 |
> | LESS (KNN-KDE) | 63.9/66.6/66.9 | 83.8/**87.8**/85.8 | 76.4/78.2/**77.9** | 61.4/**64.5**/62.3 | 20.5/31.6/34.5 |
> | LESS (UOT) | **65.3**/66.2/66.9 | 81.8/85.8/**87.8** | 76.9/77.4/77.8 | 59.1/64.3/62.3 | 14.4/33.5/35.0 |
>
> > 3. … whether the conclusions still hold when the target is a mixture of tasks….
>
> Based on your suggestion, we ran an additional experiment where the target includes a mixture of tasks with Llama 3.2 3B (see **our reply to Reviewer 8Z96**). Overall, the results are broadly consistent with the single-task selection experiment for Llama 3.2 3B in Appendix M.
>
> **References**
>
> [a] Large-Scale Data Selection for Instruction Tuning, arxiv 25.
>
> [b] Olmo 3, arxiv 25.

---

> > ### Author Rebuttal · Reviewer_KxA1 · 2026-04-02
> >
> > I would like to thank the authors for their response. According to the new results on multi-task data selection, "... LESS (RR) performs best on TyDiQA and MMLU-Pro". Meanwhile, while selecting from a dataset with longer responses (e.g., Dolci-Instruct-SFT), RDS+ (RR) is generally better. It seems that the overall picture is not fully clear.
> >
> > Additionally, I have a clarification question: what dataset does the table presented here and in the rebuttal to Reviewer 8Z96 refer to? As the tables appear identical, I assume they correspond to data selection from Dolci-Instruct-SFT.
> >
> > I will keep my score unchanged.

---

> > > ### Author Response · Authors · 2026-04-03
> > >
> > > Thank you for your thoughtful comments. Below we respond to your comments:
> > >
> > > > I would like to thank the authors for their response. According to the new results on multi-task data selection, "... LESS (RR) performs best on TyDiQA and MMLU-Pro". Meanwhile, while selecting from a dataset with longer responses (e.g., Dolci-Instruct-SFT), RDS+ (RR) is generally better. It seems that the overall picture is not fully clear.
> > >
> > > We would like to clarify the overall picture of the results. **The distance quantile experiment results are consistent across models, target tasks, and candidate pools**. In these new multi-task experiments and Dolci-Instruct-SFT experiments, similar to our previous findings, we see that only LESS (RR) shows a strong correlation between the distance quantile and loss. This highlights that longer responses present in the candidate pool might not be the only factor. However, we agree that LESS (RR) performs better for instruction selection when Tulu V2 is the candidate pool, whereas RDS+ (RR) performs better when Dolci-Instruct-SFT is the candidate pool. We are currently running experiments with additional LLMs on Dolci-Instruct-SFT to provide a more comprehensive picture. Finally, we are also looking into recent work [a] that may address the length normalization issue in LESS representations. We will include these results in the revision.
> > >
> > >
> > > > Additionally, I have a clarification question: what dataset does the table presented here and in the rebuttal to Reviewer 8Z96 refer to? As the tables appear identical, I assume they correspond to data selection from Dolci-Instruct-SFT.
> > >
> > > The table presented in the rebuttal to Reviewer 8Z96 uses Tulu V2 as the candidate pool with Llama 3.2 3B as the base model, whereas the table presented in this rebuttal uses Dolci-Instruct-SFT as the candidate pool with Olmo 3 7B as the base model.
> > >
> > >
> > > **References**
> > >
> > >
> > > [a] In Good GRACEs: Principled Teacher Selection for Knowledge Distillation, ICLR 26.

---

### Official Review · Reviewer_8Z96 · 2026-03-11

**Soundness:** 3
**Presentation:** 3
**Significance:** 2
**Originality:** 2
**Overall Recommendation:** 4
**Confidence:** 4

**Summary:**

This paper conducts a systematic evaluation of Targeted Instruction Selection. The authors perform a thorough review of the existing literature and design experiments to verify the performance of existing methods under different settings with respect to data representations, selection size, and selection algorithms. The authors also derive theorems to relate data selection and model performance.

**Compliance With Llm Reviewing Policy:**

Affirmed.

**Final Justification:**

Thank you for the detailed response. The additional experiments make the paper more compelling. I will maintain my current positive score and hope the authors can incorporate these findings in the revised version.

**Key Questions For Authors:**

1. Have the authors considered the multi-task settings, i.e., the query set and the test set are combinations of samples from multi-tasks.

2. Motivated by your Equation (1), Can we direclty optimize wasserstein distance to design selection algorithms?

**Limitations:**

yes

**Strengths And Weaknesses:**

- Soundness:​ The paper is technical sound. The authors separate data representation from the selection algorithm, making it possible to verify the contribution of each component to the final model performance. The claims in the paper are supported by extensive experimental results from performing targeted instruction selection in each setting.

- Presentation:​ The presentation is clear. The authors first show performance comparisons for each selection method under different settings, then derive two theorems to support their findings.

- Significance:​ Although the authors make some interesting claims, the paper's overall importance seems limited. First, all experiments are conducted on the Tulu dataset as the source pool. Since the paper aims to provide general findings, more datasets are needed to verify its claims, i.e., the generalizability to other source pools or targets should be tested. Secondly, the model size used in the paper is 7B. It is not clear whether model size also plays an important role in performance.

- Originality:​ Distinguishing data representation from the selection algorithm is a key conceptual novelty. While the individual components exist in prior work, the framework to isolate and study them separately is original. Besides, the authors' unification of theoretical perspectives is also a contribution to targeted instruction data selection. However, the paper does not provide a new algorithm or an improvement over existing algorithms for targeted instruction selection.

---

> ### Author Rebuttal · Authors · 2026-03-30
>
> Thank you for the thoughtful feedback on our manuscript. Below we respond to your comments:
>
> > Significance:​ …the paper's overall importance seems limited. First, all experiments ..on the Tulu … Secondly, the model size…is 7B…
>
> We thank you for your feedback. As also reflected in other reviewers’ comments, we believe the paper’s significance lies in addressing an important open question in targeted instruction selection and studying it in a **disentangled framework that cleanly isolates the contribution of each component**. In addition, we run a preliminary experiment using **Dolci-Instruct-SFT as the candidate pool** (see our reply to **Reviewer KxA1**), and we will include these results in the next version. Finally, our results (Appendix M) suggest that the relevant factor is not model size alone, but also the base model’s zero-shot performance.
>
> > Originality:​ … the paper does not provide a new algorithm or an improvement over existing algorithms …
>
> We agree that our key contribution is the disentangled framework, which enables us to critically study these selection algorithms. However, we would like to emphasize that **we also propose a new selection algorithm, Unbalanced OT (UOT) (Section 3.3)**. This algorithm explicitly solves the unbalanced optimal transport problem, allowing us to ignore outliers and samples that are less relevant to the target task. Our results in experiments (Section 5.3) show that UOT typically outperforms existing selection algorithms at larger budgets.
>
> > 1. ...the multi-task settings…
>
> Thank you for this suggestion. Based on your suggestion and **Reviewer KxA1's** suggestion, we ran an additional experiment with Llama 3.2 3B in which the query set is formed by combining samples from all five target datasets, and instructions are selected from this mixed query set. Overall, the **results are broadly consistent with those for Llama 3.2 3B in Appendix M**. First, the query loss remains strongly correlated with subset-query distance quantiles computed using LESS, with LESS (RR) achieving the strongest Spearman correlation across all five targets. Second, we find that RDS+ (RR) performs best on Codex and is generally strongest on GSM8K, whereas LESS (RR) performs best on TyDiQA and MMLU-Pro. Finally, consistent with single task results, RR tends to perform best at lower budgets, while UOT is more competitive at larger budgets. We will include the detailed plots in the final version of the paper. Below, we provide a summary of these results.
>
> **Distance Quantile Experiment.**
> | Method | BBH | Codex | GSM8K | TyDiQA | MMLU-Pro |
> |---|---|---|---|---|---|
> | RDS+ (RR) | 0.67 | 0.35 | 0.94 | 0.93 | 0.50 |
> | EMBED (RR) | 0.15 | -0.79 | 0.70 | -0.10 | 0.05 |
> | LESS (RR) | **0.96** | **0.95** | **0.99** | **0.96** | **0.99** |
>
> **Effect of Data Representation across Subset Budgets.**
> | Method | BBH | Codex | GSM8K | TyDiQA | MMLU-Pro |
> |---|---|---|---|---|---|
> | Budgets | 500/5K/10K | 500/5K/10K | 500/5K/10K | 500/5K/10K | 500/5K/10K |
> | Random | 64.6/66.5/66.9 | 85.8/**88.5**/87.2 | 76.5/75.7/73.8 | 49.1/59.7/59.8 | **35.6**/35.3/38.6 |
> | RDS+ (RR) | **65.6**/**66.9**/67.0 | **88.5**/87.2/87.2 | **78.2**/78.2/74.8 | 57.0/61.2/61.6 | 33.1/**40.4**/**40.7** |
> | EMBED (RR) | 65.4/**66.9**/**67.6** | 87.8/**88.5**/86.5 | 75.1/**78.7**/58.5 | 51.4/55.6/60.2 | 34.8/38.3/40.3 |
> | LESS (RR) | 65.0/66.8/67.4 | 79.7/86.5/**87.8** | 77.2/77.0/**77.6** | **59.8**/**63.5**/**62.9** | 22.0/35.0/35.7 |
>
>
> **Effect of Selection Algorithm across Subset Budgets**
> | Method | BBH | Codex | GSM8K | TyDiQA | MMLU-Pro |
> |---|---|---|---|---|---|
> | Budgets | 500/5K/10K | 500/5K/10K | 500/5K/10K | 500/5K/10K | 500/5K/10K |
> | LESS (RR) | 65.0/**66.8**/**67.4** | 79.7/86.5/**87.8** | 77.2/77.0/77.6 | 59.8/63.5/62.9 | **22.0**/**35.0**/**35.7** |
> | LESS (DG) | **65.3**/66.2/66.2 | 83.1/85.1/81.8 | **77.6**/**78.9**/76.9 | 59.4/62.0/62.3 | 15.6/33.9/34.9 |
> | LESS (KNN-Unif.) | 64.2/66.7/66.8 | **86.5**/**87.8**/84.5 | 77.2/76.8/77.0 | **61.6**/62.2/**63.2** | 18.4/30.3/35.0 |
> | LESS (KNN-KDE) | 63.9/66.6/66.9 | 83.8/**87.8**/85.8 | 76.4/78.2/**77.9** | 61.4/**64.5**/62.3 | 20.5/31.6/34.5 |
> | LESS (UOT) | **65.3**/66.2/66.9 | 81.8/85.8/**87.8** | 76.9/77.4/77.8 | 59.1/64.3/62.3 | 14.4/33.5/35.0 |
>
> > 2. …direclty optimize wasserstein distance …
>
> Directly optimizing for the Wasserstein distance to select subsets would **likely be intractable** because it requires combinatorial subset selection under a budget constraint, together with evaluation of a Wasserstein objective. Recent methods [b, c] rely on tractable approximations inspired by optimal transport rather than exact Wasserstein optimization. We will clarify this point in the revision.
>
> **References**
>
> [a] Large-Scale Data Selection for Instruction Tuning, arxiv 25.
>
> [b] TSDS: Data Selection for Task-Specific Model Finetuning, NeurIPS 24.
>
> [c] TAROT: Targeted Data Selection via Optimal Transport, ICML 25.

---

> > ### Author Rebuttal · Reviewer_8Z96 · 2026-04-03
> >
> > Thank you for the detailed response. The additional experiments make the paper more compelling. I will maintain my current positive score and hope the authors can incorporate these findings in the revised version.

---

> > > ### Author Response · Authors · 2026-04-03
> > >
> > > Thank you for your review and consideration. We appreciate your thoughtful comments on our work. We will incorporate these new results into the final version of the paper.

---

### Official Review · Reviewer_YcfY · 2026-03-12

**Soundness:** 2
**Presentation:** 3
**Significance:** 2
**Originality:** 2
**Overall Recommendation:** 4
**Confidence:** 4

**Summary:**

The authors tackle a very relevant problem here: how to pick the best instruction-response pairs from a massive candidate pool using just a small query set. The paper rightly points out a major pain point in the field right now, i.e., previous work on instruction selection is notoriously hard to compare because the experimental setups and metrics are usually all tangled up. To fix this, the authors propose a nice framework that splits the process into two clear parts: the data representation and the actual selection algorithm. Based on a solid set of experiments across different models and datasets, the main takeaway seems to be that gradient-based representations link up best with downstream performance. Interestingly, the results also show that as the selection budget increases, the benefits of sophisticated selection algorithms sort of fade away. I also appreciated the theoretical angle, framing these methods as minimizing the distance to the query distribution.

**Compliance With Llm Reviewing Policy:**

Affirmed.

**Final Justification:**

I find the work to be solid overall; however, the level of novelty appears to be limited. Therefore, I will maintain my original score.

**Key Questions For Authors:**

Please see my main concerns outlined in the Weaknesses section above.

**Limitations:**

Yes

**Strengths And Weaknesses:**

Strengths:
1.	The authors hit on a very real issue in the literature. Methodological inconsistencies have made it super hard to figure out what actually works, and this paper takes a good step toward fixing that.
2.	Separating the representation from the algorithm is a great move. It makes the whole experimental setup much cleaner.
3.	The empirical base is nicely done. Testing across tasks like BBH, GSM8K, TyDiQA, and MMLU-Pro gives a lot of confidence in the results.
4.	Showing that gradient-based reps consistently correlate with performance is a really helpful, practical takeaway.
5.	Unifying different algorithms under the distance-minimization perspective is a helpful way to conceptualize things.
Weaknesses:
1.	While gradient-based representations seem to be the best predictors of performance, computing them for huge candidate pools must be incredibly expensive compared to standard embeddings. I'd love to see a bit more discussion on how this computational cost actually scales. Are there any practical approximations that preserve those good predictive properties without the heavy compute?
2.	The experiments currently rely on specific pre-trained models and curated pools. I'm slightly wondering how well these trends hold up if the data distribution shifts significantly. Did the authors happen to check if the conclusions about representation quality stay consistent when using entirely different instruction corpora, or models trained on vastly different data?
3.	The authors separated algorithms and representations beautifully, but the results make it look like representation choice completely dominates anyway. This makes me wonder: does the choice of selection algorithm actually matter at all when using a really strong representation (like the gradient ones)? Or is the algorithmic effect just consistently negligible in those cases?
4.	It was honestly quite surprising to see random sampling match or even beat some of the sophisticated methods at larger budgets! That said, the paper doesn't really dig into why this is happening. Could the authors perhaps add some analysis here? For instance, does random selection become competitive because of dataset redundancy or task diversity?
5.	Finally, while the controlled experiments are great, the paper currently feels a bit light on actionable advice. It would be super helpful to give practitioners more direct guidance (e.g., propose a new framework) on how they should actually build their datasets in practice based on these findings.

---

> ### Author Rebuttal · Authors · 2026-03-30
>
> Thank you for the thoughtful feedback on our manuscript. Below we respond to your comments:
>
> > 1. While gradient-based representations seem to be the best predictors of performance..a bit more discussion on how this computational cost …scales. Are there any practical approximations…
>
> In **Appendix J** (Page 19), we provide practical approximations as we seek cheaper proxies for LESS with smaller models (135M to 3B) to train larger models (7B), thereby dramatically reducing computational costs. Our results show that using SmolLM-135M and SmolLM2-135M to produce the LESS representations to select samples to train Llama 2 7B often performs better than Random. Furthermore, using Llama 3.2 3B for targeted selection outperforms Llama 2 7B on three out of five target tasks. These results suggest that proxy models are a promising way to preserve the benefits of gradient-based representations while reducing compute, although a more thorough study of scaling and proxy-model choice remains an important direction for future work [a].
>
> > 2. The experiments currently rely on specific pre-trained models and curated pools. I'm slightly wondering how well these trends hold up if the data distribution shifts significantly. …entirely different instruction corpora, or models trained on vastly different data?
>
> Thank you for your question. We agree that our current paper does not systematically study large distribution shifts in the candidate pool. To address this, we run a preliminary experiment using **Dolci-Instruct-SFT as the candidate pool** (see our reply to **Reviewer KxA1**), and we will include these results in the final version of the paper.  On the model side, our conclusions about representation quality are already supported across multiple pretrained models (Appendix M), suggesting that the main trends are not tied to a single LLM.
>
> > 3. The authors separated algorithms and representations beautifully, but the results make it look like representation choice completely dominates anyway…
>
> Our results suggest that the choice of selection algorithm still matters even when using strong gradient-based representations, although its effect depends on the budget (**Section 5.3**). We find that LESS with greedy round-robin performs best, on average, at low budgets, whereas UOT and KNN-KDE perform better at higher budgets (Figure 7). In fact, the doubly greedy selection algorithm proposed in LESS [b] underperforms other selection algorithms on most datasets. Therefore, while representation choice is indeed the dominant factor overall, the selection algorithm can still have a meaningful effect on performance once a strong representation is fixed.
>
> > 4. It was honestly quite surprising to see random sampling match or even beat …
>
> Thank you for this helpful point. The advantage of selection algorithms is greatest at small budgets, where random subsets cover less of the data, have higher performance variance, and are less likely to intersect with the optimal subset. As the budget increases, **random subsets become broader, more stable, and more distributionally similar to the optimal subset**, which narrows the gap. To examine this empirically, we measured the Wasserstein distance between the LESS (RR) subset and a random subset on TyDiQA across different budgets, as well as the difference in query loss when trained with Llama 3.2 3B:
>
> | Budget | $W(P_{\mathrm{LESS\ (RR)}}, P_{\mathrm{Rand}})$ | $\Delta$ Query loss |
> |---:|---:|---:|
> | 500 | 0.46699077 | 1.57474783 |
> | 1000 | 0.46412995 | 1.33102079 |
> | 2500 | 0.45553076 | 1.50523953 |
> | 5000 | 0.44471049 | 0.90590242 |
> | 10000 | 0.42696017 | 0.43554587 |
>
> The monotonic decrease suggests that the random and targeted subsets become increasingly similar in distribution as the budget increases, while the difference in query loss also exhibits a near-monotonic decline.
>
>
> > 5. Finally, while the controlled experiments are great, the paper currently feels a bit light on actionable advice…
>
> We agree that the paper would benefit from more explicit guidance for practitioners. While our work argues that the claims made in existing instruction selection studies should be taken with a grain of salt, we will **revise the presentation of our findings to provide more actionable recommendations** for instruction selection (Lines 49 to 77 of the Introduction). In particular, we will clarify (1) which representations to use in different scenarios, (2) how to choose among candidate data-selection algorithms, and (3) when random sampling is sufficiently strong or even preferable.
>
> **References**
>
> [a] Small-to-Large Generalization: Training Data Influences Models Consistently Across Scale. ICLR 25.
>
> [b] LESS: Selecting Influential Data for Targeted Instruction Tuning. ICML 24.

---

> > ### Author Rebuttal · Reviewer_YcfY · 2026-04-02
> >
> > Thanks for your reponse. I will keep my score.

---

> > > ### Author Response · Authors · 2026-04-02
> > >
> > > Thank you for your review and consideration. We would be happy to answer any follow-up questions that you may have regarding our work.

---

### Decision · Program_Chairs · 2026-04-30

**Decision:**

Accept (regular)

**Comment:**

This paper looks at instruction selection for post-training for language models. Its motivation is that previous efforts at building selection techniques are difficult to use given all the different factors going on (representations of instructions, choice of algorithms, possible budgets and candidate pools, etc.) The authors seek to solve some of this by decomposing more cleanly representations versus selection approaches.

This is a solid paper with useful insights. Reviewers largely agreed with this assessment. The main concerns include questions about the choice of instruction dataset (Tulu v2), but the authors show similar results with other datasets during rebuttal. There are some questions about the connection between query loss and downstream performance, also with some reasonable answers from the authors.

Overall I view this as a nice contribution to instruction selection with valuable findings, although with some limitations.